# Genome-wide signatures of synergistic epistasis during parallel adaptation in a Baltic Sea copepod

David B. Stern [1,2 ✉], Nathan W. Anderson[1], Juanita A. Diaz [1] & Carol Eunmi Lee [1 ✉]

The role of epistasis in driving adaptation has remained an unresolved problem dating back to the Evolutionary Synthesis. In particular, whether epistatic interactions among genes could promote parallel evolution remains unexplored. To address this problem, we employ an Evolve and Resequence (E&R) experiment, using the copepod *Eurytemora affinis*, to elucidate the evolutionary genomic response to rapid salinity decline. Rapid declines in coastal salinity at high latitudes are a predicted consequence of global climate change. Based on time-resolved pooled whole-genome sequencing, we uncover a remarkably parallel, polygenic response across ten replicate selection lines, with 79.4% of selected alleles shared between lines by the tenth generation of natural selection. Using extensive computer simulations of our experiment conditions, we find that this polygenic parallelism is consistent with positive synergistic epistasis among alleles, far more so than other mechanisms tested. Our study provides experimental and theoretical support for a novel mechanism promoting repeatable polygenic adaptation, a phenomenon that may be common for selection on complex physiological traits.

[1] Department of Integrative Biology, University of Wisconsin-Madison, 430 Lincoln Drive, Birge Hall, Madison, WI 53706, USA. [2] Present address: National Biodefense Analysis and Countermeasures Center (NBACC), Operated by Battelle National Biodefense Institute (BNBI) for the U.S. Department of Homeland Security Science and Technology Directorate, Fort Detrick, MD 21702, USA. ✉email: dbstern3@wisc.edu; carollee@wisc.edu

An abundance of recent population genomic studies has found that adaptation is often highly polygenic, with hundreds or thousands of loci responding to environmental change[1–3]. A longstanding debate in evolutionary biology regards the role of epistasis in polygenic adaptation. Epistasis refers to cases where the effects of alleles at different loci are non-additive with respect to their contribution to a quantitative phenotype, such that allelic effects are dependent on the presence of other alleles at other loci[4]. Dating back to the Evolutionary Synthesis, R.A. Fisher's influential infinitesimal model assumed that many alleles contribute independent, small effects toward fitness[4–6], whereas Sewell Wright's shifting balance theory placed paramount importance on allelic combinations (i.e., epistasis)[7–9].

This debate is particularly important for our understanding of mechanisms of adaptation, including parallel evolution, and therefore the ability to predict future evolutionary genomic responses to global change[10,11]. Here we define parallel evolution as evolution driven by natural selection favoring the same loci or mutations in independent populations exposed to the same environmental challenge[12–15]. Under Fisher's model, adopted by the field of quantitative genetics, adaptation is predicted to proceed with low levels of parallelism at the genetic level where different, effectively interchangeable, loci could contribute to adaptation through a diversity of alternative evolutionary pathways[3]. A key assumption made by this model is that contributing alleles are redundant in function, resulting in the expectation of non-parallelism[16–18]. With genetic redundancy, different populations can reach the same adaptive optimum through frequency changes of different sets of alleles[3]. Alternatively, if allelic effects are non-redundant, due to epistatic effects among specific alleles, then selection might favor the same combination of alleles across independent adaptive events. In such cases when alleles are functionally linked, polygenic adaptation could become highly parallel[13,19,20].

Despite Wright's avid interest, epistasis has long been treated as statistical noise that does not contribute directly to adaptation[5,21,22]. Nevertheless, some recent theoretical studies have indicated that epistasis could have important impacts on long term responses to selection[23–25]. Synergistic fitness effects could arise among alleles (i.e., synergistic epistasis) in cases such as co-adapted gene complexes, which could be common for physiological phenotypes[26]. For example, ion transport is achieved through a suite of cooperating proteins functioning in a coordinated fashion[27,28], such that the effects of any given genetic variant are likely nonredundant and their specific effects depend strongly on the other alleles present. However, the role of epistasis remains largely overlooked in genomic studies[23,29,30] and its contribution to producing patterns of parallel polygenic evolution is unknown.

Genomic studies of adaptation in wild and experimentally evolved populations have often uncovered non-parallel molecular evolution[2,31–34], especially for small-effect loci. While some genomic studies have found more genetic parallelism than expected by chance[15,16,32,35,36], such studies tended to observe fewer than 50% of selected alleles in common between populations exposed to the same selection pressure[14]. In addition to epistasis, several factors have been proposed that could promote parallelism, including large distance to the new trait optimum, low divergence between populations, higher parallelism for large effect and high frequency alleles, pleiotropy, and others[13,37]. Yet, the relative impacts of these factors in promoting molecular parallelism remain unknown. Therefore, understanding the basis of molecular parallelism requires approaches that can accurately characterize the genomic architecture and evolutionary trajectory of polygenic adaptation to tease apart putatively important factors[38].

A limitation to identifying mechanisms of parallel polygenic adaptation has been the challenge of detecting its signatures in genomes. Polygenic adaptation is predicted to result in subtle allele frequency shifts that could be indistinguishable from genetic drift using standard approaches[6]. Analyzing replicated evolutionary events combined with whole-genome sequencing could generate sufficient power to distinguish polygenic shifts from genetic drift and uncover the degree of parallel evolution. Achieving such replication is possible with laboratory evolution experiments[38]. Yet to date, such experiments in metazoans have been limited to a small number of taxa, providing limited examples for generalization. For instance, an experimental evolution study using *Drosophila* revealed that polygenic adaptation to a novel thermal environment proceeded in a non-parallel fashion, with alleles at different loci responding to selection across independent lines[39]. However, thermal adaptation involves widespread physiological processes, with temperature affecting all metabolic pathways in ectotherms[40,41] and likely having many redundant genetic components[39]. Therefore, to determine the conditions under which polygenic adaptation can be repeatable, we require additional studies using alternate selection pressures and a variety of systems.

Thus, we employ a laboratory experimental evolution approach, combined with time-resolved whole genome sequencing (i.e., Evolve and resequence [E&R]), to dissect the genomic basis of adaptation to salinity decline in the copepod *Eurytemora affinis*, a key grazer in coastal ecosystems. Climate change is inducing rapid salinity transformations in coastal waters across the globe[42–44], yet little is known regarding the extent and evolutionary trajectory of rapid genomic adaptation to declining salinity. As such, the specific goals of this study are to (1) characterize the evolutionary genomic response of salinity adaptation in terms of the number, genomic distribution, and fitness effects of contributing alleles, (2) evaluate the degree and genetic basis of parallel evolution across replicate selection lines, given the theoretical expectations for polygenic adaptation, and (3) determine whether our experimental results are replicated in wild populations found across a salinity gradient.

To address our goals, we expose ten replicate lines to salinity decline for ten generations alongside four control lines maintained at constant salinity. These lines are then sampled for pooled whole-genome sequencing at three timepoints. Our replicated and controlled experimental design gives us the power to detect targets of selection characterized by subtle allele frequency shifts from standing genetic variants. Next, we perform computer simulations under different models of genetic architecture to evaluate the expectations for the degree of parallelism across replicate lines and explore the contribution of different factors in promoting parallel genomic evolution. Finally, we perform pooled-whole genome sequencing for eight wild populations collected from a range of salinities in the Baltic Sea to test whether the loci under selection in the laboratory experiment also exhibit signatures of selection across a natural salinity gradient in the wild.

In this study, we uncover highly parallel polygenic adaptation in a laboratory natural selection experiment. Using extensive population and quantitative genetic simulations in conjunction with our experimental results, we find strong support for a potentially widespread mechanism, namely positive epistasis, in promoting this parallel response. Our findings provide unique insights into a longstanding question in population genomics regarding the prevalence and causes of parallel evolution. In addition, our results point to the direct applicability of experimental evolution for predicting future evolutionary responses to climate change[15].

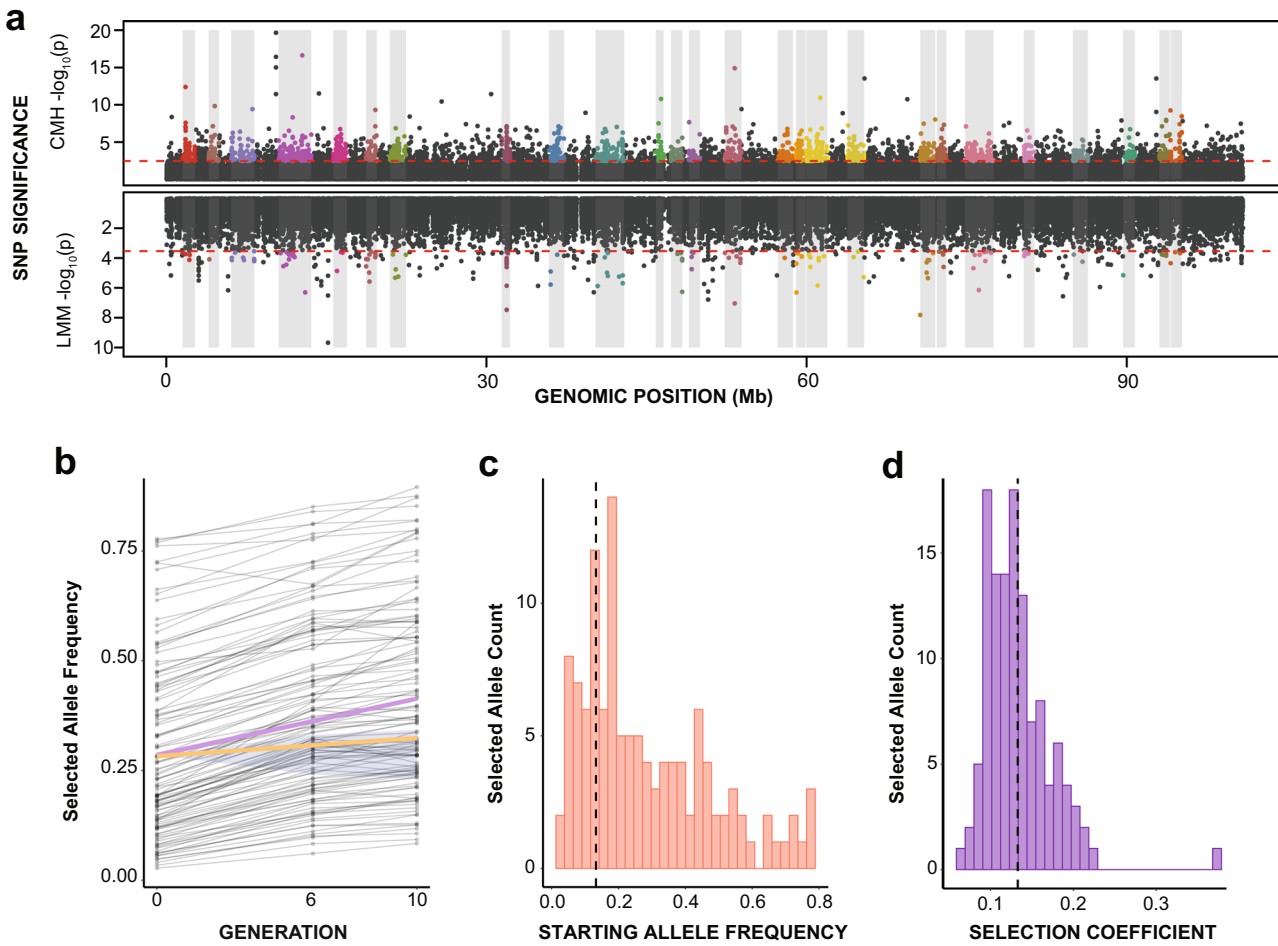

**Fig. 1 Genomic signatures of laboratory selection in response to salinity decline. a** Manhattan plot of single-nucleotide polymorphism (SNP) signatures of selection on one arbitrarily selected genomic scaffold. SNPs above the dotted red line were deemed significant (adjusted $P < 0.05$). SNPs are colored according to the selected haplotype block in which they were grouped, and shaded gray boxes delineate the 26 haplotype blocks on this scaffold. Top—Cochran-Mantel-Haezel (CMH) test for significant allele frequency changes beyond expectations from genetic drift. Bottom—linear mixed model (LMM) test to distinguish allele frequency trajectories between treatment and control lines. **b** Selected allele frequency trajectories during laboratory selection. Gray lines represent mean allele frequencies across the ten replicate lines (eight at generation ten) for each selected allele (i.e., haplotype block, N = 121). The purple line shows the average frequency for all selected alleles in the treatment lines. The yellow line shows the average frequency for all selected alleles in the control lines. The blue shaded area is the 1% and 99% quantile range of allele frequencies from 10,000 neutral simulations starting from the average starting frequency of the selected alleles. **c** Histogram of selected allele frequencies in the starting population, polarized to show the rising allele. The dotted line represents the mean frequency. **d** Histogram of selection coefficients of selected alleles, estimated using the change in frequency after ten generations across the replicate treatment lines. The dotted line represents the mean.

## Results and discussion

**The evolutionary trajectory of low-salinity adaptation**. Exposure to low salinity over ten generations resulted in a dramatic, genome-wide evolutionary response in the ten treatment (selection) lines (Fig. 1; two of the ten treatment lines went extinct between generations six and ten). The treatment lines experienced significant frequency shifts in 4,977 single-nucleotide polymorphisms (SNPs; grouped into 121 selected "haplotype blocks") spread across the genome, whereas the control lines remained relatively constant in SNP frequencies (Fig. 1a, b). To detect SNPs with signatures of selection in response to salinity decline, we performed whole-genome pooled sequencing at generations zero, six, and ten. We used Cochran–Mantel–Haenszel (CMH) tests and Chi-square tests to detect SNPs with frequency shifts that were greater than expected, relative to a model with only genetic drift, both across replicate lines and within individual replicate lines (Fig. 1a, upper panel). We also used linear mixed models (LMMs) to detect SNPs with frequency trajectories that differed significantly between treatment and control lines (Fig. 1a, lower

panel). In total, these tests uncovered 18,072 candidate SNPs with signatures of selection (out of a total of 353,188 SNPs tested).

To account for genetic linkage among SNPs, we used a recently developed approach to group candidate SNPs with signatures of selection into 121 putatively independent "haplotype blocks," consisting of 4977 proximate SNPs with correlated frequency shifts[45] (Fig. 1a, shaded boxes and colored points). The 121 selected haplotype blocks were very large in size (mean = 1.89 Mb [716 kb–6.6 Mb]; Supplementary Data 1), covering ~44% of the 518 Mb genome assembly, potentially due to strong selection pressure and few generations of recombination to disassociate linked SNPs. Patterns of allele frequency change for these haplotype blocks (hereafter referred to as "selected alleles") were based on the median SNP frequency in each line, for SNPs characterizing the selected haplotype block, following recommendations from a previous study[39].

Selected alleles in the treatment lines diverged from the starting frequency by an average of 9.9% at generation six and 12.8% at generation ten (Fig. 1b). After only ten generations, these

**Table 1 Gene paralogues with putative ion-transporter and osmoregulatory function found on haplotype blocks showing signatures of selection to salinity decline in the laboratory evolution experiment.**

| Gene Symbol[a] | Gene Description | Haplotype Block Number[b] | Starting Frequency | Selection Coefficient |
|---|---|---|---|---|
| CA-7 | Carbonic Anhydrase, paralogue 7 | 6 | 0.254 | 0.107 |
| CA-3 | Carbonic Anhydrase, paralogue 3 | 6 | 0.254 | 0.107 |
| CA-8 | Carbonic Anhydrase, paralogue 8 | 19 | 0.178 | 0.114 |
| AMT-1 | Ammonia Transporter, paralogue 1 | 43 | 0.098 | 0.137 |
| VHA-C | V-type H$^+$ ATPase, complex V1, subunit C | 43 | 0.098 | 0.137 |
| AK | Arginine kinase | 55 | 0.106 | 0.194 |
| CA-9 | Carbonic Anhydrase, paralogue 9 | 67 | 0.440 | 0.089 |
| CA-10 | Carbonic Anhydrase, paralogue 10 | 67 | 0.440 | 0.089 |
| VHA-A | V-type H$^+$ ATPase, complex V1, subunit A | 71 | 0.446 | 0.098 |
| NKA-$\alpha$−1,2,4 | Na$^+$/K$^+$-ATPase, subunit $\alpha$, paralogues 1,2,4 | 76 | 0.472 | 0.101 |
| NKA-$\beta$−2,3 | Na$^+$/K$^+$-ATPase, subunit $\beta$, paralogues 2,3 | 83 | 0.229 | 0.119 |
| CA-5 | Carbonic Anhydrase, paralogue 5 | 91 | 0.372 | 0.129 |
| Rh-1 | Rh Protein, paralogue 1 | 92 | 0.330 | 0.097 |
| VHA-G | V-type H$^+$ ATPase, complex V1, subunit G | 94 | 0.283 | 0.127 |
| NBC | Na$^+$,HCO$_3^-$ Cotransporter, paralogue unknown | 94 | 0.283 | 0.127 |
| NHA-1,2,3,4,5,6,7 | Na$^+$/H$^+$ Antiporter, paralogues 1-7 | 97 | 0.268 | 0.169 |
| AMT-7 | Ammonia Transporter, paralogue 7 | 114 | 0.206 | 0.100 |
| Rh-4 | Rh Protein, paralogue 4 | 114 | 0.206 | 0.100 |
| CA-14 | Carbonic Anhydrase, paralogue 14 | 117 | 0.378 | 0.102 |

The gene paralogues have also been implicated in previous studies of adaptation in the *E. affinis* species complex[15,28].
[a]Genes names and paralogue number are based on manual annotations of the copepod *E. affinis* complex genome[28,94].
[b]Additional details regarding the selected haplotype blocks can be found in Supplementary Data 1.

frequency shifts resulted in relatively high selection coefficients (*s*), with a mean value of 0.133 (and up to 0.372; Fig. 1d). These estimated selection coefficients were considerably larger than those estimated for temperature adaptation in laboratory *Drosophila* lines, where mean selection coefficients were only ~0.06[39,46]. Interestingly, at the start of the experiment selected alleles were found at intermediate, and often high, frequencies (Fig. 1b, c), indicating that alleles responding to selection to fresh water were often common in the saline starting population. Together, these results indicate that salinity decline elicits a rapid, polygenic evolutionary response consisting of large frequency shifts of standing genetic variants.

**Selection on ion-regulatory genes**. The evolutionary trajectory of adaptation depends on the complexity of the trait(s) under selection[2,3,6,11]. To identify potential traits under selection to declining salinity in our experiment, and determine the functional significance of selected alleles, we performed a Gene Ontology (GO) enrichment analysis for genes containing or proximal to SNPs underlying our selected alleles. This analysis implicated ion transport gene function as a primary physiological target of selection (Supplementary Data 2). Among the 91 significantly enriched GO terms (FDR-adjusted *P* < 0.05) identified using Gowinda[47], 29 (32%) were related to transmembrane transport (e.g., GO:1902476 chloride transmembrane transport), ion channel activity (e.g., GO:0005248 voltage-gated sodium channel activity), the nervous system (e.g., GO:0060079 excitatory postsynaptic potential), and muscle contraction (e.g., GO:0030432 peristalsis).

Surprisingly, genes with putative ion-transport and osmoregulatory function were found in many different selected haplotype blocks spread across the genome, suggesting possible functional coordination among distant genomic loci (Table 1). A number of these genes have previously been implicated in rapid evolution of freshwater tolerance in this species complex[15,28,48]. Of particular interest was a selected allele 2.7 Mb long that spanned a region containing seven paralogues of the *NHA* gene family, which encode sodium-hydrogen antiporter proteins (Table 1). This genomic region has been implicated in freshwater adaptation in

multiple previous studies of related *E. affinis* complex populations[15,28]. This result suggests that the genetic and evolutionary pathways that enable low-salinity adaptation are likely to be constrained.

**Exceptional genomic parallelism across laboratory lines**. We uncovered a strikingly high parallel evolutionary response among the replicate selection lines (ten lines in generation six and eight lines in generation ten), with an average of 79.5% overlap in selected alleles between replicate lines in generation ten (based on the Jaccard index[49], Fig. 2a, b, upper brown dotted line). We use the term "parallel" to refer to cases when identical alleles experienced significant frequency shifts in the same direction across the replicate selection lines. We defined a significant frequency shift in each selection line as one that was in the top 0.1% of neutral simulations by generation ten, given the allele's starting frequency (see Methods). We quantified the degree of parallelism in terms of pairwise overlap of selected alleles between replicate lines (Jaccard index[49]) and number of replicate lines in which the allele exhibited a significant frequency shift (i.e., "Replicate Frequency Spectrum"[39], Fig. 2c). Of the 121 total selected alleles (haplotype blocks), each line had significant frequency shifts for 89.0 ± 1.69 SE (74%) of the alleles at generation six and then 107.1 ± 1.70 SE (89%) alleles at generation ten. Of the alleles exhibiting significant frequency shifts, the mean pairwise Jaccard index was 0.647 at generation six and 0.795 at generation ten (Fig. 2a, b, brown dotted lines). At generation six and ten, respectively, 24 (19.8%) and 47 (38.8 %) of selected alleles exhibited significant frequency shifts in all eight surviving replicate lines (Fig. 2c, brown bars).

When simulating selection under the standard population genetics model, here called the "multiplicative fitness" model (Table 2a; Supplementary Table 1; Supplementary Fig. 1), we found that the observed degree of parallelism in our natural selection experiment had greatly exceeded expectations of this null model, given the empirical genetic architecture of the selection response (Fig. 2). Simulations of allele frequency shifts under the "multiplicative fitness" model using empirical estimates of effective population size ($N_e$) and selection coefficients (*s*), and

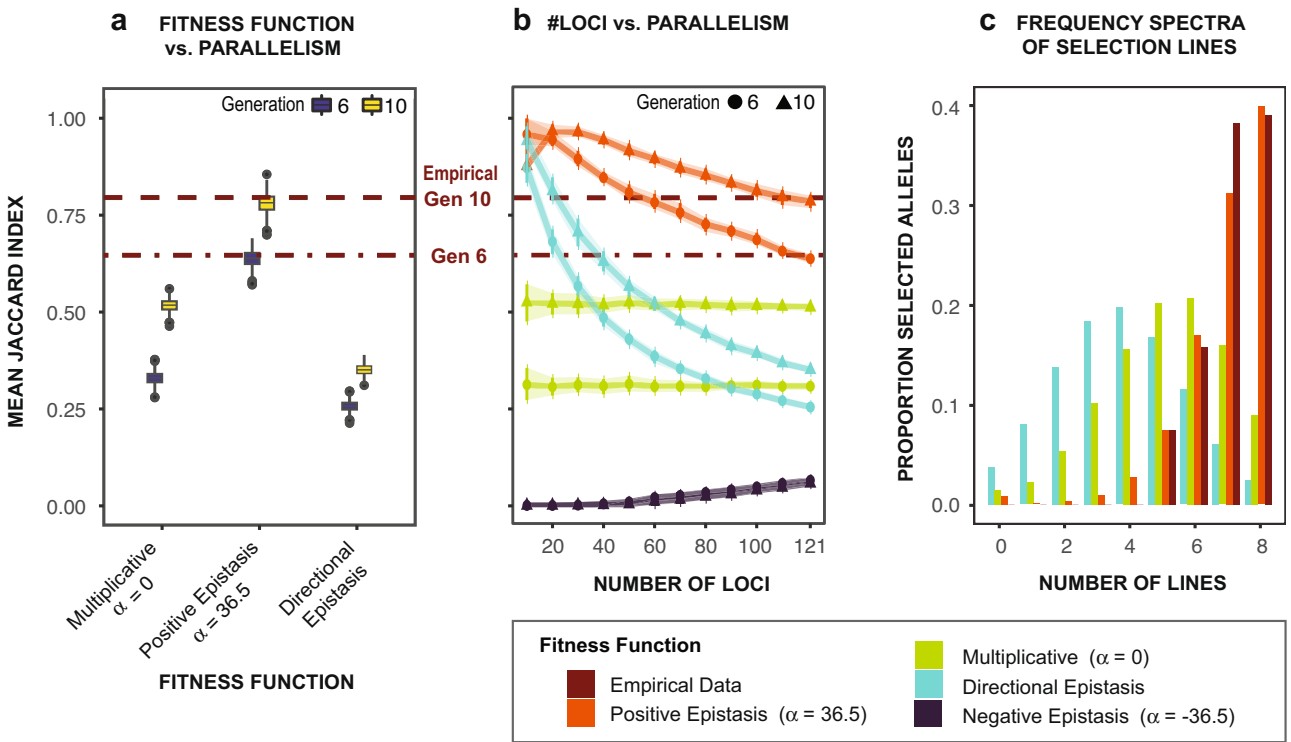

**Fig. 2 Genomic parallelism across replicate treatment lines during laboratory selection. a** Pairwise overlap (Jaccard index) of selected alleles between experimental replicate lines (dotted brown lines) relative to simulations (box plots). Box plots display median (middle line), 25th and 75th percentile (box), 5th and 95th percentile (vertical line), and data beyond the 5th and 95th percentile (single points) of mean Jaccard indices at generations six (dark blue) and ten (yellow) from 1000 simulation iterations. Empirical values most closely match those of data simulated under the "positive epistasis" model with an α parameter of 36.5 (generation six: $P = 0.69$; generation ten: $P = 0.58$), and are significantly higher than those simulated under the standard population genetic "multiplicative fitness" model ($P = 0$) and the quantitative genetic "directional epistasis" model ($P = 0$), based on $N = 1000$ simulations for each model. The "multiplicative fitness" model provides a null expectation for parallelism given the allelic selection coefficients and effective population sizes. **b** Relationship between number of loci contributing to the adaptive response and levels of parallel evolution. For 100 simulation iterations, we calculated the mean Jaccard index between simulated populations and display the mean (triangle and circle points) and interquartile range (vertical lines and shaded area) of Jaccard indices across simulation iterations. Epistatic models (Table 2), particularly quantitative fitness models, predict that the degree of parallelism among replicate lines increases as the number of loci contributing to an adaptive response declines. Levels of parallelism in our real data (horizontal dotted lines) can be replicated either by reducing the number of effective alleles in the "directional epistasis" model (teal) or increasing the α parameter of the "positive epistasis" model (orange). **c** The distribution of selected alleles in terms of the proportion of replicate lines in which the selected allele experienced a significant frequency shift at generation ten (i.e., "Replicate frequency spectrum" ref. [41]). A far greater proportion of replicate lines show higher proportions of the selected alleles in the empirical data (brown) than in the "multiplicative" (green) or "directional epistasis" (teal) model simulations, but closely match simulations under the "positive epistasis" model (orange). See Table 2 for details on the models.

---

**Table 2 Descriptions of evolutionary genetic models used in computer simulations of epistasis.**

**(a) Population genetic framework**

| Model Type | Fitness Function | Description |
|---|---|---|
| Multiplicative (no epistasis) | $\prod_{i=1}^{Nloci}(1 + h_i s_i)$ | The traditional population genetics model, in which a beneficial allele increases an individual's fitness by a fixed ratio. |
| Positive Epistasis | $\prod_{i=1}^{Nloci}(1 + h_i s_i) \cdot e^{\alpha(x-\delta)^2}; \alpha > 0$ | The effect of an allele is increased by the presence of other selected alleles. |
| Negative Epistasis | $\prod_{i=1}^{Nloci}(1 + h_i s_i) \cdot e^{\alpha(x-\delta)^2}; \alpha < 0$ | The effect of an allele is decreased by the presence of other selected alleles. |

**(b) Quantitative genetic framework**

| Model Type | Fitness Function | Description |
|---|---|---|
| Shifted Optimum Epistasis | $e^{-\frac{((x-\delta)-\mu)^2}{\sigma^2}}$ | The traditional quantitative genetics model, with a Gaussian fitness curve around a phenotypic optimum. |
| Directional Epistasis | $\left(1 + se^{r((x-\delta)+b)}\right)^{-1/s}$ | Fitness benefits saturate as the phenotype increases. |
| Truncating Epistasis | $\max\{0, 1 - e^{-a((x-\delta)+b)}\}$ | Phenotypes below a certain value are lethal. Fitness benefits saturate as the phenotype increases. |

The Phenotype: $x = \frac{\prod_{i=1}^{Nloci} h_i s_i}{\prod_{i=1}^{Nloci} s_i}$

Horizontal Shift: $\delta = \overline{phenotype_{simulation\ initial}} - \overline{phenotype_{baseline}}$

$h_i = \begin{cases} 0 & \text{If the individual carries 0 copies of the allele at locus } i \\ 1/2 & \text{If the individual carries 1 copy of the allele at locus } i \\ 1 & \text{If the individual carries 2 copies of the allele at locus } i \end{cases}$

with significant frequency shifts determined in the same manner as for the empirical data, resulted in mean Jaccard indices of only $0.33 \pm 0.015$ SD at generation six and $0.52 \pm 0.016$ SD at generation ten (Fig. 2a, left). These values were significantly lower than the Jaccard indices based on our real data (Fig. 2a, brown dotted lines; $P = 0$, $N = 1000$ simulations). In addition, our empirical Replicate Frequency Spectrum (Fig. 2c, brown bars) was markedly different from that obtained under the multiplicative model (Fig. 2c, green bars), with a much greater proportion of selected alleles responding in most of the replicate lines in our empirical data. Allele frequency trajectories under the standard multiplicative fitness model are independent with respect to individual fitness and depend on the selection coefficient and population size. Therefore, non-parallelism could arise due to genetic drift and random sampling[35,50,51]. As our simulations used accurate estimates of effective population sizes, starting allele frequencies, and selection coefficients (see Methods), the much greater degree of parallelism of allele frequency shifts in our experimental lines was quite exceptional (Fig. 2).

Physical linkage could theoretically result in a high degree of parallelism between lines if selected alleles tend to be found close together on the same haplotype and are inherited together[52]. Alternatively, if alleles tended to be unlinked, competition between alleles at different loci (i.e., Hill-Robertson effects) could reduce parallelism[53]. Although our selected alleles were putatively unlinked, we also explored the effect of physical linkage between alleles by simulating realistic recombination architectures for the 121 alleles (haplotype blocks). Surprisingly, we found that in fact physical linkage only slightly decreased the Jaccard index by ~0.003 on average across models (Supplementary Fig. 2). Even with reduced recombination rates, the effect of physical linkage on parallelism was very small (approximately 0.005 decrease in Jaccard index; Supplementary Fig. 2), likely due to the large genomic distances between selected loci and the large effective population sizes in each of the replicate lines (mean $N_e = \sim1750$; see Methods). As populations in our simulations started from linkage equilibrium, future work should investigate the effect of starting LD structure on the degree of parallelism.

Overall, our results differed sharply from other studies of polygenic adaptation. For example, Barghi et al.[39] examined selection response to a changing temperature regime, which affects a broad range of physiological processes[40,41], and found highly heterogeneous responses among replicate experimental lines with much lower parallelism than predicted under the "multiplicative fitness" model. The exceptionally high levels of parallelism found in our experiment, relative to the baseline multiplicative model (Fig. 2), indicate that a factor beyond the allele frequencies, heterogeneous effect sizes, small divergence among populations, and population size drove parallel evolutionary responses across our replicate lines (see next sections for further discussion).

**Positive epistasis promotes rapid parallel evolution.** Despite Sewall Wright's interest in the role of epistasis in promoting adaptive evolution[7–9], the classical quantitative genetics literature has often treated epistasis as a statistical residual noise component of phenotypic variance[4,23,29,54,55]. In this study, we found that positive synergistic epistasis was likely a major force in driving parallel selection in the replicate experimental lines (Fig. 2). For instance, our simulations indicate that positive epistasis could account for the 0.27 difference in Jaccard index between our empirical data and simulations of the "multiplicative fitness" model at generation ten (Fig. 2a, see more details below). Based on our finding of ion uptake as the primary functional trait under selection in our experiment (e.g., Table 1; Supplementary

Data 2), we hypothesized that synergistic effects among alleles could have promoted the high levels of parallel evolution observed between selection lines. Like ion uptake, many physiological traits with complex architectures could be primarily controlled by coordinated protein networks with extensive interacting effects[26]. Ion transport is known to be driven by the coordinated activity of a suite of primary and secondary ion transporters[27,56], several of which were identified here as targets of selection (Table 1). Such functional coordination could result in combinations of functionally linked alleles that are favored by selection. If the positive fitness effects of favorable alleles were enhanced by the presence of other favorable alleles (i.e., synergistic epistasis), then the set of favored alleles would rise together in frequency in a highly parallel fashion across replicate experimental lines.

When we tested whether our empirical results were consistent with this hypothesis using theoretical simulations, we found that positive epistatic effects among alleles of selected loci greatly increased the degree of parallelism in simulated data (Fig. 2; Supplementary Figs. 2–5). We performed extensive Wright-Fisher simulations under different models of epistasis to determine whether, and what mode of, epistasis could explain the high degree of parallelism in our data (Table 2; Supplementary Fig. 1). We simulated both population genetic and quantitative genetic characterizations of epistasis that all allowed allelic fitness effects to depend on the presence of other selected alleles in an individual. The population genetic models of epistasis were an extension of the "multiplicative" model, but where fitness effects are an exponential function of the sum of the allelic effect sizes in an individual[57], including either positive or negative epistatic interactions (Table 2a). The quantitative genetic models of epistasis were constructed to explore differently shaped fitness functions and were parameterized to allow allelic effects to increase with the addition of other selected alleles in the early stages of adaptation (Table 2b). These simulations were constructed either to model selection on 121 haplotype blocks or the 4977 SNPs underlying those haplotype blocks using the effect sizes, population sizes, and linkage structure from our real data.

We found that simulations of the population genetic "positive epistasis" model most closely matched the levels of parallelism in our real data (Fig. 2a, middle; Fig. 2b, c, orange). In contrast, the quantitative genetic epistasis models were unable to produce levels of parallelism close to our real data (average Jaccard index across quantitative genetic models in generation ten $= 0.38 \pm 0.03$ SD), even though our simulations imposed a strong selection pressure to maximize possible parallelism (see Methods; Supplementary Fig. 1). An important parameter of the population genetic "positive epistasis" model is the $\alpha$ parameter (Table 2a), which describes the increase in fitness benefits of gaining more beneficial alleles at other loci, i.e., the strength of the epistatic effect (see Methods for a description of this parameter). Therefore, we used Approximate Bayesian Computation (ABC) to estimate the value of this parameter that could reproduce the observed mean Jaccard index among replicate selection lines in our empirical data. We found that the ABC-estimated $\alpha$ value of 36.5 produced Jaccard index values that closely matched our real data at both generations six ($P = 0.69$, $N = 1000$ simulations) and ten ($P = 0.58$, $N = 1000$ simulations), indicating that this level of positive epistasis was sufficient to explain the degree of parallelism observed in our data. By comparing the Jaccard index under simulations of the "multiplicative" model ($\alpha = 0$) relative to the "positive epistasis" model ($\alpha = 36.5$), we found that positive epistasis could account for an increase of up to 0.27 in Jaccard index of selected alleles among the replicate lines at generation ten (Fig. 2a).

To further evaluate the strength of epistasis in terms of the effective number of loci under selection, we simulated selection using a variable number of loci contributing to the adaptive response under different models of epistasis (Table 2). Under the quantitative genetic models of adaptation used here, parallelism increases with fewer contributing loci (Fig. 2b, teal lines), as independent populations would then have fewer possible genomic routes to adaptation and each allele would have a larger relative effect on the phenotype[3]. If low-salinity adaptation is achieved through frequency changes of alleles with coordinated functions (i.e., as in the case of synergistic epistasis), then these functionally linked alleles would behave like a smaller number of alleles contributing to the adaptive response and would tend to have a parallel selection response. We found that depending on the fitness function, our 121 selected haplotype blocks (alleles) responded as if they were 20 or fewer alleles in terms of the degree of parallelism (Fig. 2b, intersection between teal lines and our empirical data; Supplementary Fig. 4). This result implies that despite the many unlinked loci involved, the selection response might consist of a small number of co-adapted gene complexes.

This study is novel in demonstrating that positive epistasis can promote parallelism on a polygenic, genome-wide scale. Previous studies have shown that epistasis can limit evolutionary trajectories, and thereby promote parallelism among closely related populations, but only at the scale of a few loci[20,37]. By combining experimental evolution and extensive genetic simulations in this study, we were able to identify a mechanism that could drive parallel evolution across many loci. Indeed, positive epistatic interactions could actually increase the rate of evolutionary responses[23] and therefore be prevalent during rapid adaptation. Future studies of physiological adaptation should take into account the potential for allelic coordination to impact rates and patterns of adaptation[30].

**Selection on high-frequency alleles alone cannot explain the high levels of parallelism.** As our previous study using this copepod system implicated selection on high-frequency alleles, potentially maintained by balancing selection, as a promoter of parallelism[15], we sought to determine the extent to which this factor could also explain the high levels of parallelism observed in our experiment in this study. We found that the effect of starting frequency on the Jaccard index was much lower than the estimated effect of epistasis (see next paragraph). We found that SNPs characterizing selected haplotypes were at significantly higher starting frequencies than non-selected SNPs (mean minor allele frequency [MAF] of 0.139 vs. 0.089, respectively; two-sided Kolmogorov-Smirnov test, $D = 0.214$, $P = 0$). This finding was not an artifact of selected SNPs with low starting frequencies being difficult to detect using the CMH test[58]. In fact, our CMH test results showed a negative correlation between the test statistic and starting frequency (Pearson's $r = -0.118$, $P < 0.0001$), suggesting that lower-frequency SNPs were in fact more detectable as targets of selection. Our LMM test showed no significant correlation between the test statistic and starting frequency (Pearson's $r = 0.03$, $P = 0.345$), indicating no clear bias.

Interestingly, higher frequency alleles did not exhibit a more parallel response. We found that starting MAF was somewhat negatively correlated with the number of replicate lines in which the allele experienced a significant frequency shift (Pearson's correlation test–Generation six: $r = -0.260$, $t$ value $= -2.94$, DF $= 119$, $P = 0.00399$; Generation ten: $r = -0.214$, $t$ value $= -2.40$, DF $= 119$, $P = 0.0183$). We also used theoretical model simulations to evaluate the potential contribution of selection on high-frequency alleles toward promoting parallelism. Using the same simulation schemes as above, we compared the degree of parallelism observed

when (1) alleles started from mutation-drift balance (i.e., neutrality) and (2) alleles started with a frequency of 0.5, meant to maximize the potential effect. We found that simulating selection from mutation-drift balance only decreased the Jaccard index by ~0.001 on average across models, relative to simulations from empirical frequencies (Supplementary Fig. 3). Simulating from frequencies of 0.5 increased the Jaccard index by ~0.05 on average across models. Overall, this effect of starting frequency on the Jaccard index was much lower than the estimated effect of epistasis (which showed a Jaccard index increase of up to 0.27).

The elevated frequencies for selected alleles were consistent with the presence of synergistic epistasis, given that epistasis could aid in maintaining balanced polymorphisms[55]. For instance, our previous study found that many SNPs with signatures of parallel directional selection during freshwater invasions by North American *E. affinis* complex populations exhibited signatures of ancient balancing selection in the native range populations[15]. This result indicated that balancing selection could maintain many alleles that are beneficial in fresh water at elevated frequencies in the saline native populations. Synergistic epistasis acting among those particular beneficial alleles could help explain the effectiveness of balancing selection at maintaining genetic variation at many sites across the genome over long periods of time[55].

**Experiment-selected loci are also under selection across salinity gradients in the wild.** Population genomic analyses revealed that experiment-selected alleles also exhibited signatures of selection in wild *E. affinis* populations from eight Baltic Sea locations (Fig. 3a) spanning a salinity gradient (2.5–18.7 practical salinity units [PSU]; Supplementary Data 3). This striking result indicated that the repeatability of the selection response extends beyond the laboratory and was recapitulated in natural populations. In fact, most SNPs underlying experiment-selected alleles ($N = 3655$; 74%) harbored segregating SNPs in the wild populations. Due to the polygenic nature of the selection response observed in the laboratory, we used the method of Berg and Coop (2014)[59] to test for a population genomic signature of polygenic selection on the SNPs underlying experiment-selected alleles (haplotype blocks) in the Baltic Sea populations. Rather than analyzing each SNP separately, this method tests a set of trait-associated loci for a signature of selection, captured in the $Q_X$ statistic, based on the degree to which SNPs at those loci vary across populations and covary with each other. We found that the $Q_X$ statistic for experiment-selected loci was significantly higher than that of the null distribution of genomic background loci matched in minor allele frequency (experiment-selected loci $Q_X = 22.46$, background loci mean $Q_X = 9.61$, $P = 0.016$; Fig. 3b), indicating a strong signature of selection and high variance in experiment-selected SNP frequencies among populations. We also used this method to calculate polygenic scores for freshwater survival for each Baltic Sea population using selection coefficients for each SNP as a proxy for its effect size. Interestingly, polygenic scores were significantly negatively correlated with mean annual salinity ($\beta = -0.064$, $P = 0.021$), indicating that alleles that responded to selection in our experiment were more common in Baltic Sea populations from lower-salinity habitats.

Minor allele frequencies for experiment-selected loci were significantly higher than non-selected loci in every sampled wild population (e.g., Fig. 3c; Supplementary Data 4). The fact that this pattern was observed across populations inhabiting very different annual salinities points to spatially varying selection potentially maintaining important SNPs at high and intermediate frequencies, provided there is sufficient gene flow among populations. Using the $f_4$ statistic[60], we found genomic evidence for

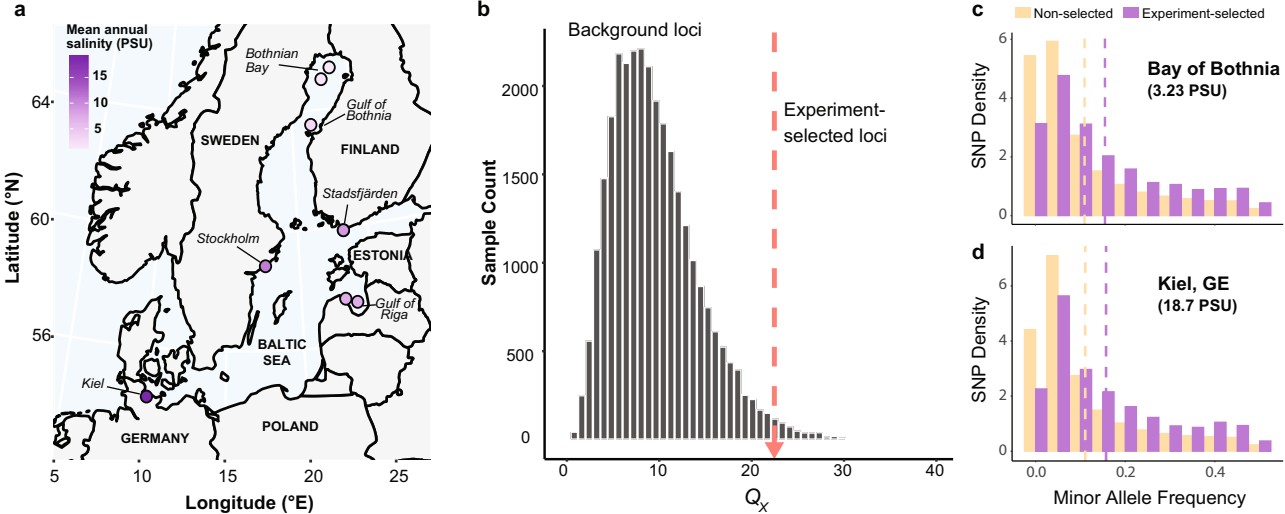

**Fig. 3 Experiment-selected loci also exhibit signatures of selection in the wild populations from the Baltic Sea. a** Locations sampled for pooled genomic sequencing across the Baltic Sea. Sampling sites are colored according to their mean annual salinity estimated from the International Council for the Exploration of the Sea (Supplementary Data 3). The map was generated using data from https://www.naturalearthdata.com (Accessed June 2020). **b** SNPs underlying experiment-selected alleles harbor a strong signature of polygenic selection across the wild populations, as captured by the $Q_X$ statistic. The $Q_X$ statistic measures how quickly the experiment-selected loci evolve and how strongly SNP frequencies covary across wild populations. The empirical $Q_X$ estimate for the experiment-selected SNPs (red dotted line) is far greater than the null distribution (gray histogram) based on the neutral population history and genomic background. **c, d** Minor allele frequencies (i.e., folded allele frequencies) for SNPs underlying experiment-selected alleles (purple) are significantly higher than non-selected SNPs (yellow) in two geographically divergent Baltic Sea populations from highly different salinity conditions. Vertical dotted lines display mean minor allele frequency for each category of SNP.

considerable gene flow among geographically distant populations. The $f_4$ statistic tests whether the possible phylogenetic trees relating populations are consistent with admixture among populations. Gene flow or admixture is inferred if all three possible trees for a group of four populations are rejected by the data[60]. Testing all 126 possible groupings of four populations, 47 groupings (37.3%) showed significant support for gene flow (Supplementary Data 5), including populations from the most distant sampling sites in the northern and southern regions of the Baltic Sea.

Baltic Sea habitats are characterized by considerable depth and latitudinal gradients in salinity[61,62]. Such spatial heterogeneity of selection pressures could result in the maintenance of adaptive genetic variation, given the right combination of genetic architecture, selection pressures, population sizes, and migration rates[63–67]. Indeed, characteristics of *E. affinis* complex populations might enable local adaptation with gene flow to serve as a key force maintaining adaptive genetic variation. These features include large effective population size[68,69] and negative genetic correlations between high and low salinity tolerance[70–72]. Furthermore, spatially varying selection could favor the synergistic effects among adaptive alleles, as such interactions would produce greater variance in fitness and faster adaptation to local conditions[23].

The extent to which the same alleles that responded to selection in the laboratory also exhibited significant signatures of natural selection in wild populations was striking. An ongoing question is whether laboratory evolution experiments can truly replicate natural processes and inform predictions of evolutionary responses in the wild, given the simplified conditions. The advantages of laboratory evolution experiments are the ability to isolate a selective pressure and to have complete knowledge of the study populations' history during the experiment. While numerous experimental studies have shed important insight into the genomics of adaptation[73,74], it remains unclear whether these results can be translated to natural populations[75]. Our results

suggest that we can expect the loci detected in our experiment to play an important role in future responses to salinity decline in the Baltic Sea and other high latitude regions.

## Concluding remarks

Polygenic adaptation is predicted to proceed in a non-parallel manner when a trait optimum can be achieved through frequency changes in different combinations of effectively interchangeable (redundant) loci[3]. In this study, we found that polygenic adaptation proceeded in a highly parallel, repeatable manner (Figs. 1, 2), in stark contrast to what has been found in some comparable experimental evolution studies[39,76]. It is possible that the low levels of parallelism often reported in wild genomic studies[2,31–34] could be due, in part, to a lack of power in identifying the targets of selection. In this study, we found parallelism not only among replicate laboratory lines, but also between the laboratory and wild populations. Our computer simulations allowed us to rule out the impacts of allelic effect sizes, starting frequencies, population sizes and divergence, and physical linkage on producing the observed high degree of parallelism. The primary mechanism we identified as promoting this polygenic parallelism, namely positive epistasis, may be widespread particularly for physiological adaptation. A few genomic studies of physiological adaptation have reported instances of molecular and phenotypic parallelism, but did not compare support for different mechanisms underlying parallel evolution[77–79]. Assessing the prevalence of genome-wide epistasis and its impact on parallel evolution across a diversity of traits and systems would provide insights into the role of positive epistasis in promoting rapid and repeated adaptation.

Among the most ecologically and economically impactful consequences of global climate change is the rapid change in ocean salinity throughout the globe[80]. Due to massive increases in ice melt and precipitation, salinity is predicted to decline at unprecedented rates in higher latitude coastal waters, by up to 5 PSU in the coming decades[42–44]. Salinity is arguably the strongest driver of aquatic biogeographic distributions[81–83]. Therefore,

changes in this critical environmental stressor will force populations to adapt, migrate, or face extinction. Zooplankton populations, which constitute the largest animal biomass in water column habitats, may be particularly vulnerable to changes in salinity because of their limited ability to migrate and often narrow salinity tolerance ranges[70,72,84]. This study makes important contributions toward understanding the evolutionary mechanisms of how populations will respond to such rapid changes in environmental salinity.

Our study uncovered genomic and experimental evidence that *E. affinis* populations in the Baltic Sea have a high probability of adapting rapidly to decreasing salinity caused by climate change. The observation that the same alleles responded to laboratory selection across replicate lines suggests that the ability to adapt to decreasing salinity relies on a predictable set of genetic variants that we also found present in the wild populations. Although our laboratory selection experiment was performed on a population derived from a single location, potentially biasing our discovery of selected loci, we did find that the majority of experiment-selected genetic variants (74%) were segregating in wild Baltic Sea populations and that many were found at high and intermediate frequencies (Fig. 3). These results imply that the standing genetic variation required for adaptation to freshening water is both available and abundant across a broad geographic range of *E. affinis* populations in the Baltic Sea. Moreover, epistatic interactions among the beneficial alleles could allow adaptation to be rapid and repeatable. Such evolutionary resilience for this copepod, which serves as a critical food source for many important fisheries, could be vitally important for maintaining healthy ecosystems in the face of climate change.

## Methods

**Population sampling, design of the laboratory evolution experiment, and sequence data collection**. All copepod populations examined in this study, including that used in the laboratory evolution experiment and those surveyed in the wild were considered *Eurytemora affinis* proper, which is the European clade of the *Eurytemora affinis* species complex[85–87]. The *E. affinis* copepods used in the laboratory natural selection experiment were collected from Kiel Canal in Kiel, Germany (latitude = 54°19′ 59.88″N, longitude = 10°9′0″) in 2017 (~1000 copepods) and on May 30, 2018 (85 gravid females and 40 juveniles). The two collections of copepods were mixed and maintained at 15 PSU to increase population size and acclimate to laboratory conditions. Two samples of adult copepods (25 male and 25 female each) from the mixed culture were collected for pooled whole-genome sequencing (Pool-seq) to represent the starting population for the laboratory natural selection experiment and capture variance in starting SNP frequency. The culture was then split into 14 equally sized beakers. Control lines (N = 4) were maintained for the duration of the experiment in 15 PSU water made with Instant Ocean, along with Primaxin (20 mg/L) to avoid bacterial infection. The control lines were fed the marine alga *Rhodomonas salina* every three to four days with water changed weekly. The ten treatment lines were exposed to decreasing salinity over the first six generations until they reached 0 PSU (Lake Michigan water, ~300 μS/cm conductivity), and then maintained at 0 PSU for four additional generations.

Beginning at generation two, salinity declination proceeded at each generation as follows: 5 PSU, 1 PSU, 0.1 PSU, 0.01 PSU, 0 PSU. Animals were not transferred individually to the next generation but instead were allowed to survive and reproduce undisturbed with overlapping generations. The generation number was monitored by assuming a generation time of approximately three weeks[72,88]. Treatment lines were fed a 50:50 mixture of *R. salina* and the freshwater alga *R. minuta* at 5 PSU and only *R. minuta* at 1 PSU and below. These algal species are closely-related cryptophytes[89] that are both rich in long-chain polyunsaturated acids and are the preferred food source of *Eurytemora* populations[90,91]. As these algal cells are highly sensitive to osmotic shock during salinity change, changing the food source from saline to freshwater *Rhodomonas* was unavoidable during salinity decline in the experiment. Although population size remained fairly constant for most treatment lines, two treatment lines went extinct between generations six and ten (BSE-7 and BSE-10) and were therefore sequenced only at generation six.

Individual adult copepods (N = 50; 25 male and 25 female) were collected for sequencing from each laboratory selection line at generations six (after one generation at 0 PSU in the treatment lines) and ten (after five generations at 0 PSU in the treatment lines). Two of the control lines (BS3C and BS4C) were also sampled at generation 20 to increase the number of sampled timepoints for our LMM test for selection. Sampled copepods from each line (Supplementary Data 6)

were pooled and their DNA was extracted using the DNeasy Blood and Tissue Extraction kit (Qiagen, Inc.). Paired-end whole-genome sequencing libraries were prepared using the Nextera DNA kit (Illumina Inc.) and sequenced on four lanes of Illumina Hi-Seq 4000 and one lane of Illumina NovaSeq 6000 at the University of Chicago Genomics Facility, generating an average of ~117 million paired-end (100 bp) reads per pool.

Additionally, wild *E. affinis* populations were collected from eight locations in the Baltic Sea (Fig. 3; Supplementary Data 3) using bongo and WP2 nets with 100 μm mesh. These sampling locations spanned a range of mean annual salinities from low (~3 PSU) to higher salinity (~19 PSU). Mean annual salinity for each site was calculated from the International Council for the Exploration of the Sea (ICES) database (https://ocean.ices.dk/Helcom/Helcom.aspx?Mode=1; accessed June 2020), using data collected from 1995-2020. From each population, individual copepods (ranging from 50 to 200 in number) were pooled and their DNA was extracted using the DNeasy Blood and Tissue Extraction kit (Qiagen, Inc.). Paired-end whole-genome sequencing libraries were prepared using the Illumina Nextera DNA kit (Illumina, Inc.) and sequenced on five lanes of an Illumina HiSeq 4000 sequencer at the University of Chicago Genomics Facility, generating an average of ~176 million paired-end (100 bp) reads per pool.

**Reference genome assembly and SNP calling**. A draft genome for *Eurytemora affinis* complex was constructed from long-read and long-range sequencing technology. To generate genomic data for assembly, an inbred line was generated from 30 generations of full-sibling mating of copepods derived from a saline population in Baie de L'Isle Verte, St. Lawrence marsh, Quebec, Canada (Atlantic clade, aka *E. carolleae*[86]). Prior to DNA extraction, the culture was treated with a series of antibiotics to reduce bacterial contamination, including Primaxin (20 mg/l), Voriconazole (0.5 mg/l for at least 2 weeks prior to DNA extraction), D-amino acids to reduce biofilm (10 mM D-methionine, D-tryptophan, D-leucine, and 5 mM D-tyrosine, for at least for 2 weeks prior to DNA extraction). DNA was extracted from pooled adult copepods and used to generate Pacific Biosciences (PacBio) long-read data (2.65 million subreads, 30.2 Gb, read N50 = 19.25 kb), Dovetail Chicago (213 million 150 bp read pairs) and Hi-C (171 million 150 bp read pairs) Illumina HiSeq X data.

PacBio reads were assembled into contigs using *wtdbg* v2.5[92] and polished with Racon v1.4.3[93]. This procedure resulted in an assembly of size 575 Mb in 3013 contigs with an N50 of 951 kb, an N90 of 106 kb, and maximum contig length of 7.7 Mb. Contigs were scaffolded with HiRise™ (Dovetail Genomics) using Dovetail Chicago® and Hi-C data. This procedure resulted in a highly contiguous assembly with 90% of the genome present in four scaffolds (1706 total scaffolds, 0.02% gaps), likely corresponding to the four chromosomes of *E. affinis* observed in karyotype (data not shown). These four scaffolds consisted of 518 Mb of sequence. Gene annotations were generated by mapping annotations from the previously published, short-read-assembled genome of *E. affinis* complex[94] using LiftOff v1.6.1[95].

European *E. affinis* populations, the subject of the laboratory selection experiment, are highly genetically divergent from North American populations of the *E. affinis* complex[85,86], from which the draft genome was derived. Therefore, to maximize the number and accuracy of SNP calls, a pseudo-reference genome was assembled for SNP calling following an approach used in a previous study[96]. This approach uses a reference transcriptome as an anchor to assemble Pool-seq data around coding regions, generating a reference assembly that includes coding sequences and surrounding sequence with putative regulatory function.

To generate RNA-seq data for the reference transcriptome, approximately 100 individuals of all life stages were collected from the European *E. affinis* laboratory culture derived from Kiel, Germany. Animals were pooled and subjected to a Trizol/Qiagen RNeasy hybrid total RNA extraction protocol. An mRNA sequencing library was prepared using the Illumina TruSeq Stranded mRNA kit and sequenced on an Illumina HiSeq 4000 sequencer, generating approximately 186 million 100 bp paired-end reads. Raw RNA-seq reads were processed using BBDuk in the BBtools v38.37 package (https://sourceforge.net/projects/bbmap/) to filter adapter sequences, low complexity sequences, and low quality (Q < 10) bases using a sliding window (ktrim = r k = 23 mink = 11 hdist = 1 qtrim = w trimq = 10 minlen = 36 entropy = 0.01 entropywindow = 50 entropyk = 5 tbo).

To filter sequences that may have originated from microbial contamination prior to assembly, reads were mapped against a database of reference and representative bacterial, archaeal, and fungal genomes from the NCBI RefSeq using Bowtie 2 v2.3.5 and were removed if they mapped concordantly[97]. Cleaned reads were assembled using Trinity v2.6.6 in strand-specific mode[98]. mRNA expression estimates were made using RSEM v1.3.1[99] with cleaned reads mapped to the Trinity assembly with Bowtie 2[97]. Only the most highly expressed Trinity "isoform" per 'gene' was retained to obtain the most highly supported transcriptome assembly. Transcript sequences were clustered to 95% similarity using CD-HIT v4.7[100] to reduce redundancy associated with allelic variation and assembly errors. Transdecoder v5.5[101] was used to predict open reading frames and coding sequences. Predictions included BLASTP[102] (BLAST + v2.7.1) hits to annotated proteins of the *E. affinis* complex draft genome and HMMER v3.2.1 (http://hmmer.org) hits to the Pfam protein database[103].

The resulting non-redundant transcriptome assembly was used to anchor the de novo assembly of Pool-seq data around coding regions to obtain a pseudo-

reference genome. First, the "left" pairs of Pool-seq reads from the starting laboratory population were mapped to the coding sequences of the transcriptome with BWA-MEM v0.7.17[104], retaining reads with a mapping quality >20. The corresponding "right" pairs of the mapped reads were extracted and the mapped reads were assembled using Trinity v. 2.6.6[98], as Trinity was developed to assemble highly variable sequences with potentially uneven coverage. The resulting assembly was clustered to 95% similarity using CD-HIT v4.7[100]. The aforementioned mapping and assembly procedures were repeated one additional time, mapping reads to the Trinity-assembled Pool-seq data rather than to the transcriptome. In this way, the assembly was extended into genomic regions surrounding the transcriptome-derived coding sequences. Finally, only assembled contigs with a significant BLASTN hit (E value < 0.001) to the draft genome (see above) were retained to further eliminate assembly errors. The pseudo-reference genome ultimately spanned a greater portion of the genome with greater contiguity than the transcriptome (Supplementary Data 7). These pseudo-reference genome contigs were arranged and oriented into scaffolds using BLASTN hits (E-value < 0.001) of the contigs to the long-read genome assembly and retaining the top hit per contig.

Raw Pool-seq reads collected from the laboratory selection lines and Baltic Sea wild populations were processed with Trimmomatic v0.39[105] to filter adapter sequences and low-quality bases using a sliding window (LEADING:3 TRAILING:3 SLIDINGWINDOW:4:15 MINLEN:36). Quality-filtered Pool-seq reads were mapped to the pseudo-reference genome using BWA-MEM v0.7.17[104] retaining only reads that mapped concordantly with a mapping quality >20. Duplicate reads were removed using Picard v2.18.27 (http://broadinstitute.github.io/picard) and regions around indels were realigned using GATK v3.8[106]. For the laboratory lines and wild populations separately, SAMtools v1.3.1 was used to convert BAM files into mpileup format after removing low-quality alignments and bases ($Q < 20$). VarScan v2.4.3[107] mpileup2cns was used to call SNPs for each of the mpileup files using the following options: "–min-coverage 20 –min-avg-qual 20 –min-var-freq 0.0001 –variants –output-vcf". The resultant VCF files were processed using the R package *poolfstat* v. 1.1.1[108] retaining bi-allelic SNPs with 4 reads required for a base call, an overall minimum MAF of 0.01, and a minimum and maximum coverage of 10 and 200 reads, respectively. This procedure resulted in 367,846 SNPs called in the laboratory samples and 693,438 SNPs called in the wild samples.

**Estimating effective population size.** We used the change in SNP frequencies over time to estimate the effective population sizes ($N_e$) for each line with the R package *poolSeq* v0.3.5[109,110] and WFABC v1.1[111]. Using the *poolSeq* model, we made estimates using both the "Plan I" (census-size dependent) and "Plan II" (census-size independent) methods, and, given the uncertainty in the census population size, assumed a range of reasonable census sizes for the "Plan I" calculations. We made $N_e$ estimates in genomic windows of 1000 SNPs in size and took the median of those windows (Supplementary Data 8).

**Detecting signatures of selection in the laboratory natural selection experiment.** Multiple approaches were used to test for signatures of natural selection imposed by decreasing salinity in our SNP frequency data. We used a version of the CMH test that uses allele frequency data from multiple timepoints, and also accounts for overdispersion due to genetic drift and Pool-seq sampling, to detect SNPs with frequencies that changed more than expected under random genetic drift across lines[112]. We also performed a similar Chi-square test to detect line-specific signatures of selection using the same R package. These tests were performed considering only the eight treatment lines with three sampling time-points, using the mean SNP frequency from the two starting laboratory population samples as the generation zero frequency for each line. The CMH and Chi-square test statistics were calculated using estimated effective population sizes for each line to calibrate *p* values accounting for genetic drift (Supplementary Data 8). *P* values were converted to *q*-values to correct for multiple testing using the R package *qvalue*[113] and SNPs with a *q* value < 0.05 were considered significant.

As the CMH and Chi-square tests did not take full advantage of our sampling design with control and treatment lines, SNPs were also tested for signatures of selection using LMMs in the R package *lme4*[114], considering data from all control and treatment lines. The goal of this test was to detect SNPs with frequency trajectories that were significantly different between treatment and control lines. Therefore, we could detect SNPs with deterministic signatures of directional selection to decreasing salinity, while accounting for the effects of random genetic drift and selection to laboratory conditions. SNP frequency change ($x_D - x_A$) was considered the response variable in the linear regressions with weights proportional to sequencing depth and number of individuals sampled ($N_{eff}$)[115]. An angular transformation (i.e., arcsine square root transformation) was applied to SNP frequencies to normalize the percentage values and reduce bias due to different starting frequencies[116–118].

Likelihood ratio tests (LRT) were used to test for the significance of the effect of *treatment* (2 levels) on SNP frequency change from the start of the experiment (the response), considering the fixed effect of *generation* (2 levels) and the random effect of *line* (14 levels). The following two models were compared: (1) $y_i \sim generation + (1|line) + \varepsilon_i$ and (2) $y_i \sim generation + treatment + generation{:}treatment + (1|line) + \varepsilon_i$, where $y_i$ is the SNP frequency change from the starting population at the $i$th SNP and $\varepsilon_i$ is the Gaussian error at the $i$th SNP, given the sequencing coverage and number

of individuals sampled. LRT test statistics were compared to the Chi-square distribution with two degrees of freedom to obtain *p* values. *P* values were converted to *q* values to correct for multiple testing[113] and SNPs with a *q* value < 0.05 were considered significant.

To account for linkage among SNPs, which could inflate signatures of selection for neutral SNPs, putatively independent selected "haplotype blocks" were identified using the R package *haplovalidate*[45,119] considering the eight treatment lines with three sampled timepoints. This approach identifies and delimits selected haplotype blocks by clustering candidate SNPs with correlated frequency changes. The union of candidate SNPs from all three tests for selection (CMH test, line-specific Chi-square, and LMM; $N = 18{,}072$) were used as input to *haplovalidate*, which was run with default settings. Identified selected haplotype blocks were considered as selected alleles. The median frequency of SNPs characterizing each haplotype block was used to estimate allele frequencies at each timepoint in each line. To determine whether an allele had responded to selection in each line and timepoint, we simulated 10,000 iterations of neutral evolution using the R package *poolSeq*[109] for each allele's starting frequency using an effective population size of 1750 (the approximate average of $N_e$ estimates across lines and methods; Supplementary Data 8). Alleles were then considered to be under selection in each line and timepoint if the frequency had increased more than the 99.9th percentile of neutral simulations at generation ten.

Selection coefficients ($s$) were estimated by linear modeling with *lme4*[114] from the slope of the linear regression of logit-transformed allele frequencies against *generation,* averaged across lines. The slope of this regression has been shown to accurately estimate $s$, assuming a continuous-time approximation to the Wright-Fisher model and codominance[109]. The factor *line* was considered a random effect in the linear model with weights proportional to sequencing depth and number of individuals sampled ($N_{eff}$)[115]. We also calculated $s$ using the *poolSeq* R package, which also uses *lme4* to regress logit-transformed allele frequencies against generation, but without considering the random effect of *line* or variable sequencing depth. The two estimates of $s$ were well correlated (Pearson's correlation test, $r = 0.79$, DF = 119, $t$ value = 14.12, $P < 0.001$).

**Simulations of laboratory selection.** To evaluate the expected degree of parallelism under different genetic architectures (Table 2; Supplementary Fig. 1), we performed extensive Wright-Fisher simulations using SLiM v3.7[120]. We included both population genetic and quantitative genetic characterizations of epistasis (Table 2; Supplementary Fig. 1; Supplementary Table 1) to test which could best explain the extent of parallelism in our experimental data. We simulated ten replicate lines under selection with a constant population size of 1750 individuals, based on the average effective population size estimated across lines and methods (Supplementary Data 8). Mirroring the design of our E&R experiment, we recorded allele frequencies of selected loci in every population at generations zero (starting allele frequencies), six, and ten. To introduce the additional variance in allele frequencies due to population sampling and sequencing, we sampled 100 individuals from each population to be included in each "pool", and then sampled allele copies from the pool according to empirical sequencing coverage of each locus.

We accounted for directional (positive or negative) epistasis under a population genetic (multiplicative fitness) framework by extending Keightley and Otto's[57] characterization of epistasis (Table 2a). The "multiplicative fitness" model is the standard population genetic model in which allelic effects are independent with respect to individual fitness and allele frequency trajectories depend on the selection coefficient and effective population size (Table 2a, first equation). In this population genetic framework, epistasis is described as an exponential function of the square of the sum of the effect sizes of the beneficial alleles an individual carries, where the α parameter (β in Keightly and Otto[57]) indicates the strength and sign of the epistatic effect. Furthermore, this framework allows for mutations of variable effect sizes and for mutations with larger individual effect sizes to have larger epistatic effects (Table 2a, second and third equations).

In addition, we included several quantitative fitness functions with epistatic effects, in which the positive fitness effects of alleles increase in the presence of additional alleles (Table 2b; Supplementary Fig. 1). The quantitative trait (QT, quantitative trait) models were chosen to replicate the models used in similar simulation studies of replicated evolution experiments[38,50] and are also available in the simulation software *MimicrEE2*[121]. We simulated under multiple different quantitative fitness functions for two reasons: (1) our experiment allowed individuals to survive and replicate naturally (i.e., we did not select and transfer individuals to the next generation), and therefore the exact mode of selection is unknown, and (2) our experimental design included a moving optimum (salinity declined over generations 2–6), which was not perfectly captured by available fitness functions. Therefore, simulating under a range of different quantitative fitness functions could help capture the allele frequency dynamics under the quantitative genetic paradigm, regardless of the exact (and unknown) mode of selection.

The Shifted Optimum QT model is a standard QTL model with a Gaussian fitness curve around a phenotypic optimum (Table 2b; Supplementary Fig. 1). The Directional model is similar to the Shifted Optimum model with the "stabilizing" component removed, such that fitness benefits plateau with increasing phenotype instead of a fitness reduction. The Truncating model is very similar to the Directional model, but where having a low phenotype is lethal rather than low

fitness. Our "Truncating" model is equivalent to the "diminishing returns epistasis" model in *MimicrEE2*[121]; however, we changed the name to avoid confusion, because negative epistasis also has diminishing fitness effects as the phenotype increases.

Overall, parameter values under the QT models (Supplementary Table 1) were chosen to allow allelic effects to increase as the mean population fitness increased in the early stages of adaptation (i.e., positive epistasis). Parameter values for the Truncating model were chosen such that ~25% of the starting population were culled and fitness benefits saturated quickly above that point. The Directional model parameters were chosen to closely match the "scale" of the fitness benefit saturation. The parameters of the Shifted Optimum model were chosen such that the slope of the fitness functions matched the Directional model and the fitness optimum occurred roughly where the fitness benefits saturate. Trait optima were chosen such that the distribution of population phenotypes had just reached the trait optimum by generation ten (examples of initial and final population phenotype distributions are shown by the horizontal colored bars in Supplementary Fig. 1). Therefore, populations did not spend time in a 'non-adaptive' phase and our expected parallelism under this model was not reduced. Despite this fact, the levels of parallelism observed under a trait optimum model (quantitative trait models) did not approach the observed empirical levels of parallelism (Supplementary Figs. 2–4).

Simulations were conducted by modeling: (1) 121 unlinked alleles (haplotype blocks), (2) 121 alleles with linkage and recombination, and (3) 4,977 SNPs on 121 unlinked haplotype blocks with linkage and recombination between SNPs on the same haplotype block. In simulations with linkage, alleles recombined based on their physical distance using the copepod *Tigriopus californicus* recombination rate of 1.6 cM/Mb[122]. Selected alleles started at observed starting frequencies unless otherwise specified. In all cases, whether an individual carried an allele at the start of the simulation was based solely on that allele's starting frequency (e.g., if an allele has frequency $p$, an individual carries no copies with probability $(1 - p)^2$ and one copy with probability $2p(1 - p)$).

Unless otherwise specified, simulations were carried out with parameter values shown in Supplementary Table 1. Simulations of the QT models with 121 haplotype blocks used some different parameters from simulations of 4977 SNPs due to the differences in range of starting phenotypes in the two scenarios (Supplementary Fig. 1, top versus bottom rows, horizontal gray bars). 100 iterations were run for each simulation in which the number of selected loci was varied (e.g., Fig. 2b). 1000 iterations were run for each simulation in which the shape of the fitness function was varied (e.g., Fig. 2a, c), and the mean Jaccard index between populations was recorded for each iteration. As with the empirical data, simulated alleles were considered under selection in a given line and timepoint if their frequency change was greater than 99.9% of neutral simulations performed using the R packages *poolSeq*[109].

To estimate the α parameter of the positive epistasis model and determine whether the positive epistasis model could recreate the observed levels of parallelism, we performed Approximate Bayesian Computation using the R package *EasyABC* v1.5[123] and the simulations as above using SLiM 3. The mean empirical Jaccard index at generations six and ten was used as target summary statistics. The Lenormand sequential algorithm[124] was used to estimate α using a uniform prior distribution [0,50]. 1000 post-burn-in iterations were used to summarize the posterior distribution of α with the weighted mean taken as the point estimate.

**Explanation of the *a* parameter of the population genetic epistasis model.** In our population genetic model of epistasis (both positive and negative), the *a* parameter signifies the strength of epistasis. That is, this value represents how much the fitness effect of one locus is affected by the genotype at other loci. For simplicity, consider a haploid individual with $N$ biallelic loci where the beneficial alleles have an equal effect size $s$. Note that under these assumptions, our population genetic epistasis models (Table 2) simplify to:

$$(1 + s)^n \cdot e^{a\left(\frac{x}{N}\right)^2} \qquad (1)$$

Under a standard population genetic, multiplicative framework (Table 2) an individual with $n$ beneficial alleles ($w_n$) will have a fitness $1 + s$ times greater than an individual with $n - 1$ beneficial alleles ($w_{n-1}$), regardless of the value of $n$. In contrast, under our model of epistasis, $w_n$ will be

$$\frac{w_n}{w_{n-1}} = (1 + s) \cdot e^{a\frac{2n-1}{N^2}} \qquad (2)$$

times greater than $w_{n-1}$. The exponential term is equal to 1 if $a = 0$ and grows larger (with $a > 0$) with increasing $n$. Thus, the fitness benefit of gaining a beneficial allele increases with the total number of beneficial alleles that individual carries when $a > 0$.

For a concrete example, imagine a haploid individual with 121 loci and $a = 36.5$, our empirical ABC estimate. Under a multiplicative model, each copy of the allele increases an individual's fitness by $(1 + s)$ times. On the other hand, under our epistatic model the fitness benefit of the having a single allele is

$$\frac{w_1}{w_0} = (1 + s) \cdot e^{\frac{36.5}{121^2}} \approx 1.002(1 + s), \qquad (3)$$

which is slightly more than is expected under the multiplicative model. In contrast, the fitness benefit of having 121 alleles over having 120 alleles is

$$\frac{w_{121}}{w_{120}} = (1 + s) \cdot e^{\frac{36.5 \cdot 241}{121^2}} \approx 1.823(1 + s), \qquad (4)$$

which is much greater than expected under the multiplicative model. In other words, the fitness benefit of having more beneficial alleles is increased by the presence of beneficial alleles at other loci. $a < 0$ has the opposite effect, such that the fitness benefit of beneficial alleles is decreased by the presence of beneficial alleles at other loci. The strength of epistasis (either positive or negative) increases with increasing $|a|$.

**Simulating selection under different starting allele frequencies.** To explore the effect of starting allele frequencies on genetic parallelism, we repeated our simulations with populations initialized using neutral allele frequencies (drawn from the SNPs without significant signatures of selection using any of the three tests [Methods—*Detecting signatures of selection in the laboratory natural selection experiment*]), and where every locus had a starting frequency of 0.5. In the cases where the number of loci contributing to a trait was varied (Fig. 2b, Supplementary Fig. 5), the loci included were randomly selected from the total available for each simulation. For SNP simulations, the positions of the selected SNPs were retained. As our SNP simulations considered SNPs on the same haplotype blocks as linked, it is possible that if, for instance, 100 loci contributed to adaptation, all 100 could be located on the same haplotype block or they could all be located on their own haplotype block and be unlinked.

Varying the initial allele frequencies or number of loci could lead to some issues with the choice of parameters for the fitness functions (e.g., if all loci selected have very low starting frequency, every individual's phenotype could fail to meet the truncation threshold). For this reason we introduced $\delta$, a "horizontal shift" parameter, defined to be the difference between the mean initial phenotype in the current simulation ($\overline{phenotype_{simulation\ initial}}$) and the mean initial phenotype in the simulations using the empirical starting frequencies ($\overline{phenotype_{baseline}}$). The horizontal shift parameter adjusts the initial phenotype distribution so that it matches the initial phenotype distribution of the simulations using the empirical starting allele frequencies and number of loci.

**Gene annotations and functional enrichment.** Gene ontology enrichment analyses were performed using Gowinda v1.12[47], which takes into account gene length and SNP density, using all SNPs underlying selected haplotypes. Genes within 10 kb of a selected SNP were included in the analysis, with each gene counted only once ("gene" mode). Gene ontology terms (Biological Process, Molecular Function, and Cellular Component) associated with each gene model were derived from significant ($E < 0.001$) BLASTN hits to the UniProtKB Swiss-Prot database.

**Detecting signatures of selection in wild Baltic Sea populations.** The method of Berg and Coop[59] was used to test for signatures of polygenic selection on experiment-selected SNPs in eight wild populations from the Baltic Sea (Fig. 3a; Supplementary Data 3). Using this method, a "polygenic score" for freshwater survival was calculated for each population by taking the sum of each experiment-selected SNP's frequency multiplied by its effect size. Estimated selection coefficients from the laboratory experiment were used as proxies for effect size for freshwater survival. The $Q_X$ statistic, which measures the variance in polygenic scores considering the neutral population history, was calculated to test whether experiment-selected SNPs exhibited signatures of selection. The neutral population history (i.e., SNP variance-covariance matrix) was estimated using three samples of 50,000 background SNPs. To test whether experiment-selected loci evolved faster than background loci, that is, exhibited a signature of selection, a null distribution of $Q_X$ statistics was calculated using 10,000 samples of background SNPs matched in minor allele frequency to the experiment-selected loci. This null distribution was used to calculate a p value for whether the experiment-selected loci evolved significantly faster than the genomic background.

To test for signatures of gene flow among geographically distant populations, we calculated the $f_4$ statistic[60] implemented in TreeMix v. 1.13[125] for every combination of four populations, calculating standard errors with groups of ten SNPs. We considered the $f_4$ test significant if all three of the possible four-population trees were rejected with a |Z-score| > 2.

**Reporting summary.** Further information on research design is available in the Nature Research Reporting Summary linked to this article.

## Data availability

The raw Pool-seq data generated in this study have been deposited to the NCBI Short Read Archive under BioProject ID PRJNA844002. The *E. affinis* complex (Atlantic clade) draft genome and the Baltic pseudo-reference genome are available on Dryad (https://doi.org/10.5061/dryad.r7sqv9sdz)[126]. Study information has been deposited to BCO-DMO (https://www.bco-dmo.org/project/816918). Allele frequency data (SNP and haplotype block) are available on https://github.com/TheDBStern/Baltic_Lab_Wild

(v.0.0.1[127]). Data used in this study from publicly available databases include the International Council for the Exploration of the Sea (ICES) database (https://ocean.ices.dk/Helcom/Helcom.aspx?Mode=1), NCBI RefSeq (https://www.ncbi.nlm.nih.gov/refseq/), the Pfam database (https://pfam.xfam.org/), and the UniProtKB Swiss-Prot database (https://www.uniprot.org/uniprot/?query=reviewed:yes).

## Code availability

Custom analysis and simulation scripts used throughout this study are available on https://github.com/TheDBStern/Baltic_Lab_Wild (v.0.0.1[127]).

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

## Acknowledgements
This work was supported by the National Science Foundation OCE-1658517, NSF DEB-2055356, and French National Research Agency ANR-19-MPGA-0004 to C.E.L., the Michael Guyer Postdoctoral Fellowship and the NIH/NHGRI Genome Sciences Post-doctoral Traineeship (NIH-5T32HG002760) to D.B.S., and UW-Madison Department of Integrative Biology Graduate Summer Research Awards to J.A.D. and N.W.A., and UW-Madison Graduate Recruitment Fellowship to J.A.D. The authors would like to thank the scientists and crew members of the F.S. Alkor and the R.V. Aranda, especially Jan Dierking, Thorsten Reusch, and Pekka Kotilainen, for their help collecting *E. affinis* from the Baltic Sea and for their generosity in allowing us to join their excursions on the F.S. Alkor and the R.V. Aranda. Samples from Stockholm and Kiel were collected by Per Hedberg and Fabian Wendt, respectively. The Finnish samples were collected with the help of Jonna Engström-Öst. Thanks to Ziting Zhang and Nick Mathers for their work maintaining the laboratory lines. Jesse Weber, Nathaniel Sharp, Christopher McAllester, Teresa Popp, and Benny Kleinerman provided helpful discussion and comments on the manuscript. Computation was performed using the computing resources and assistance of the UW-Madison Center for High Throughput Computing (CHTC) in the Department of Computer Sciences.

## Author contributions
D.B.S., C.E.L., and N.W.A. contributed to the design of the study. D.B.S., J.A.D., and C.E.L. performed copepod collections from the Baltic Sea. D.B.S. and C.E.L. performed and monitored the laboratory natural selection experiment. D.B.S. and J.A.D. performed molecular laboratory work and analyzed the raw sequence data. D.B.S. and N.W.A. executed the genetic simulations and statistical analyses. All authors wrote and approved the final manuscript.

## Competing interests
The authors declare no competing interests.
