## [Peer Review File · Nature Communications]

Genome-wide signatures of synergistic epistasis during parallel adaptation in a Baltic Sea copepodReviewers' Comments:

Reviewer #1:

Remarks to the Author:

This paper by Stern et al. adds an exciting new approach to the previous work done in the *Eurytemora affinis* copepod species on adaptation to changing environmental conditions. The use of replicated selection lines combined with a survey of genomic variation in natural populations of this species in the Baltic region make this a valuable contribution. I think that the implications for evolutionary understanding and insights into how species can handle changing environmental conditions will make this interesting to a broader audience. Overall, I found the paper well-written with most of the analyses well-explained and appropriately interpreted. I do have a few concerns that could impact the framing of the broader message about the importance of epistasis that I think should be addressed and some other questions described below as well.

Polygenic model and epistasis: I think the findings for the models of polygenic loci that could explain patterns of parallelism are interesting but it is not clear to me that this clearly implicates epistatic interactions. My reading of this section is that they are adapting the polygenic model by just reducing the number of loci until they find the signature of parallel evolution that they find in their dataset. Are there other ways of doing modeling that more directly incorporate epistatic effects and how its existence would directly impact patterns of parallel evolution? In line 347 they mention that models are being developed that do incorporate epistasis but it doesn't seem like these are the models that they used. If the 30 gene model of polygenic inheritance fits that pattern of parallel evolution does that necessarily imply that widespread epistasis is the best explanation for why they see so many more SNPs under selection? Could a long history of balanced polymorphism across a large number of loci without epistasis explain the observed patterns (maybe via increased linkage disequilibrium by tending to be in the same individuals due to a history of environmental selection but not via physical linkage)?

Detection of parallelism—it isn't clear to me exactly how the selected SNPs are determined; can a SNP be found significant for selection in only one line of the experiment or does it depend on repeated selection across the lines. If it is the latter this would seem to bias it towards detecting the cases of parallel selection.

Physical linkage and inversions—If the genome assembly is rather fragmented and overall patterns of physical linkage are harder to determine, it seems like this weakens some of the arguments to a degree. For the analysis of lack of mapping of one read of a paired end read, have other studies shown this to be an effective way of looking for inversions in this situation? It seems plausible, but there could also be a lot of other artifacts and other genome assembly issues that could make it less likely to work.

Line 146 The statement 'help resolve' seems a bit strong—maybe something like contribute to our understanding? Similarly in Line 154, the 'with great precision' feels a bit strong as well.

Figure 1a: how do you get the allele frequencies averaging ~ 0.5 when presumably natural variation was of all frequencies at the start? Maybe it is just that the axis label is confusing as it sounds like you did a regression to get these values (and it is not raw changes in allele frequencies plotted).

Reviewer #2:

Remarks to the Author:

In manuscript 'Parallel polygenic adaptation driven by epistasis in laboratory and wild populations of a Baltic Sea copepod' authors have evolved copepods in low salinity environment and investigated the genomic signatures of selection. They have performed computer simulations mimicking selective

sweep and polygenic adaptation of quantitative traits and concluded that synergistic epistasis is prevalent in their study. Although authors emphasize that epistasis is the reason the empirical data doesn't fit the simulated evolutionary scenarios, epistasis is not explicitly modeled and simulated; the lack of fit of empirical data to the simulated ones has been interpreted as epistasis. The authors are correct that the role of epistasis in adaptation is not well studied, but modeling epistasis is very difficult as the information about the functional network of genes in many sexual eukaryotes is still missing. But I'm afraid that the analysis and simulations of the manuscript doesn't fully support the role of epistasis in adaptation in this study system. There are also some aspects of data analysis and simulations that need further investigation.

Major comments:

1. There is a general mis-understanding/presentation of polygenic adaptation (e.g. L139); That, polygenic adaptation would not result in parallel response. Many factors including distance to the new trait optimum, heterogeneity of alleles (effect size and starting frequencies), epistasis, pleiotropy, and GxE affect polygenic adaptation, and in conditions genomic response can be parallel under polygenic adaptation. If the new trait optimum is far from the starting phenotype of population or few adaptive alleles are present or effect size of alleles are large, the genomic response of polygenic adaptation in fact will be parallel and quite similar to selective sweep. The tone of discussion in this manuscript, however, is that all previous theoretical or empirical studies which have shown or suggested non-parallel response across replicate populations have missed the point, or overlooked the true signal due to the noise in dataset. Many aspects of data analysis and simulations should be done more thoroughly to make sure that the observed pattern is true.

Moreover, the authors don't show or even model epistasis in their simulation, they show that the observed parallelism in the empirical data fits a model with fewer alleles of larger effect size. The parsimonious explanation/interpretation could be that the number of adaptive alleles in this study has been overestimated. If authors believe that their findings support prevalence of epistasis in polygenic adaptation, epistasis should be modeled explicitly. In line 89-92, it is mentioned that the loci with epistasis are functionally linked and their interactions are not additive. While in the simulations, effects are additive and no functional linkage (or even physical linkage) is considered.

2. Authors state that parallelism and synergistic epistasis in this study are consistent with a complex physiological trait (L45-47), but in Discussion and GO term analysis section (L400-402) where the authors claim to have identified the true functional/selected trait, it is mentioned that the studied trait (adaptation to low salinity) is not as complex as other traits such as adaptation to temperature and it's simpler. So if the studied trait is not as complex as temperature adaptation, then the trait might be oligogenic with few contributing loci, and the true number of selected loci might be less.

3. In Fig. 2, the authors show that the parallelism in their study is even higher than sweep, while sweep is conditional on fixation of all alleles, I am curious to see what is the interpretation that the experiment has more parallelism than selective sweep model would predict? If the parallelism in empirical data is similar to simulations of 30 loci, how can the authors reject the hypothesis that the true number of selected targets is less than the estimated number of selected SNPs (1156)? Considering that linkage has not been taken into account in this experiment which has small population size and probably high drift, and selection should be estimated using a more suitable software, and both number of loci and selection coefficient are important parameters of simulations, further investigation of these factors is needed (see below).

Data analysis:

4. L617: The simulations that are used to estimate the number of loci varies both the number of loci and their effect sizes. In fact, the results show that simulations with fewer loci and different effect size fits their empirical data better. Authors should perform additional analysis (looking for selected haplotype blocks rather than assuming independence among selected SNPs) to account for linkage among the selected SNPs (which might reduce the number of selected sites) and use other tools to estimate selection coefficient (which affects the trajectory of alleles). These 2 parameters determine the allele trajectory and level of parallelism among simulated replicates, which heavily affects the conclusions of this paper (see below).

Are the selected SNPs localized in specific regions of genome? Give the small population size and high effect of drift, could the selected SNPs be linked and not independent? If plotted in a Manhattan plot, do the selected SNPs form peaks (linked selection) or are they scattered across the genome? There are many tools developed to reconstruct the partial selected haplotype block using time series data. I suggest authors use one of the tools to account for the effect of linkage in the number of selected SNPs.

5. L581: There are many tools developed specifically for estimation of selection coefficient. A recent benchmarking study (<https://link.springer.com/article/10.1186/s13059-019-1770-8>) has shown that CLEAR provide the most accurate estimation of s . I suggest authors use one of these established methods and compare their estimated s .

6. The CMH test identifies consistent allele frequency changes across majority of replicates, so if an allele is selected in few replicates only, it would have high p-value in the CMH test. Previous studies that have been cited in this manuscript have used Chi-square or Fisher's exact test to identify replicate-specific selection (Barghi et al. 2019, Otte et al. 2020). It would be good if authors perform additional analysis to identify replicate-specific response, if present.

7. The authors have used linear regression to contrast selected and control lines. If CMH test is used to contrast control and selected lines, will the same SNPs be identified as outliers?

Simulations:

8. The population size seems to be 500, how can the estimated N_e be 2000, 4 times higher than the census population size? This parameter plays a very important role in simulations, as 2000 is a large N_e and will result in low drift and high parallelism. PoolSeq package is used for N_e estimation, it would be good if authors specify which functions/parameters were used (Couldn't find any relevant command in the github)? N_e is normally computed in windows along the genome, is the reported N_e the median of all windows? Are low frequency SNPs filtered, since these might inflate the estimated N_e ?

The authors perform simulations mimicking experimental design of Tobler et al. (2014) and their study to show that their results are not biased by linkage and the number of identified selected SNPs are not inflated by long-range LD. N_e play an important role in these simulations. Authors should clarify the estimated N_e (see below).

9. The authors have looked at decay of LD (L202) and concluded that their study doesn't suffer from the effect of linked selection/linkage. But the decay of LD in Tobler et al. study is quite similar. A follow-up study on Tobler study on the same dataset has shown that despite the quick decay of LD at 200bp, long-range LD can still be prevalent, not due to segregating inversions but frequency increase of haplotype blocks (Franssen et al. 2014). I suggest in order to have a more accurate estimation of selected sites (which are quite important for the simulations) effects of linkage should be taken into account.

10. Authors have used MimicEE2 for simulations to assess the effect of linkage but for simulations of evolutionary scenarios they have used other tools that don't account for linkage. Authors should account for linkage for their sweep (and polygenic adaptation) simulations. I suggest using MimicEE2/SLiM or any other simulation tools for all simulations specifically to account for linkage among selected loci.

11. L522-525: The simulated polygenic adaptation is a 'quantitative trait after a shift in the trait optimum'. However, the selection regime in this study doesn't fit this description as the salinity of environment (the selective factor) has changed from generation 2-6. It is not obvious from the description of experiment whether only the surviving individuals were transferred to the next generation (truncating selection) or not (moving trait optimum). But the selection mode in the simulations of polygenic adaptation should be modified to fit the experiment.

12. For simulations of polygenic adaptation why 0.9 is chosen as the new optimum? how far is this new optimum from the phenotype of starting population in the simulations? Would authors observe similar results if they change the distance to trait optimum?
13. 10 iterations are too few, at least 500-1000 simulations should be performed.
14. The estimated Jaccard index for simulations is pooled for all 10 iterations. To properly mimic the experimental design, it should be computed for each set of 10 replicates, and the mean and SD across total iterations be reported.
15. If the distance to trait optimum is the same in polygenic adaptation simulations for 1156 and 30 loci, and in each simulation loci have different effect sizes (i.e. each of 30 loci have higher effect size than 1156 loci), it might take much longer for the population with 1156 loci to reach the trait optimum, while in the simulation with 30 loci the population might have reached the new trait optimum after 10 generations. In addition to trajectories of allele frequencies, the phenotype of populations should also be checked to make sure populations in both simulations have reached the trait optimum, otherwise it's like comparison of apple and oranges.
16. L267-269 (Supplementary figure 4) why 0.1 frequency change is used as threshold for selected alleles? the expected frequency change depends on the starting frequency of alleles. This threshold should be adapted depending on the starting frequency of alleles.
17. The details of parameters used for the simulations should be explicitly specified not just described.
Minor comments:
18. L73-75: There are many natural systems, for example sticklebacks, that show highly parallel response, mostly involving large effect alleles.
19. L132: It would be helpful if authors elaborate how references 35 and 36 suggest the prevalence of epistasis in this study system.
20. L144-146: Parallel polygenic adaptation has been repeatedly observed in many organisms, in natural and experimental observations. The most famous experimental populations is study of longevity in *Drosophila melanogaster* evolved over 800 generations with many replicates (<https://academic.oup.com/mbe/article/34/4/831/2897202>).
21. Is there any phenotypic information about these selected replicates? Have they converged phenotypically?
22. L527: The diet of selected populations has changed in the course of selection experiment. What is the effect of diet on the genomic response? It would be good to clarify why the diet has been changed for readers who are not familiar with this study system.
23. L558: what is the correlation of SNP frequencies between the two starting population samples?
24. Which time point was used for the CMH test, generation 6 or 10?
25. The authors compare the similarity of replicates using Jaccard index but they often refer to it as 'pairwise overlap', I suggest they use the conventional term 'Jaccard index' and cite the original paper.
26. L646: DAVID doesn't account for gene length which affects the enrichment analysis, it would be goof to use a software that takes gene length into account.
27. Fig. 1a: are the trajectories of the same SNPs shown for both selected and control lines? if yes,

why the starting frequencies differ among these lines?

28. Fig. 1d. I don't think s should be computed/reported for non-selected SNPs!

29. Fig. 2. The summary statistics in Fig. 2c is Replicate frequency spectrum, it would be good to use the proper name and cite the original paper.

30. L181: how is the frequency change of 0.1 consistent with polygenic adaptation, the reference is not related to this threshold.

31. L364-366: what is the direct evidence that the effect of epistasis is more prevalent than starting frequency of allele?

32. L389-391: are these GO terms significantly enriched?

33. Fig. 3c and d. Is the MAF folded? if not, how the derived allele was determined?

Reviewer #3:

Remarks to the Author:

In this study, the authors investigate adaptation of the copepod *Eurytemora affinis* to low salinity environments, using both laboratory selection experiments and population genomic studies of natural populations. They report evidence for highly polygenic adaptation, with dozens to hundreds of loci under selection. Parallelism is much more frequent than expected under standard models of polygenic adaptation; the authors argue that widespread positive epistasis could explain this finding.

The experimental design and dataset are very impressive. However, I am concerned about several aspects of the analysis, and thus with the resulting conclusions.

First, I am concerned about the authors' treatment of linkage disequilibrium (LD). LD could in principle account for several notable aspects of the data, including the seemingly large number of loci under selection (since some of these loci could simply be hitchhiking with selected sites), and the high level of parallelism (since linked sites would be expected to show parallel trajectories). The authors are well aware of these issues, and undertake extensive simulations in an attempt to address them. A few concerns here:

(1) In the simulations, the authors use an N_e of 2000 (lines 195; 602-603; 611). I don't see this number justified anywhere - where does it come from? 2000 strikes me as very high - census size is ~ 500 , and N_e is typically less than census size, often by a lot. In this case, the authors will have overestimated N_e by an order of magnitude or more, and the population recombination rate for the simulations will be off by the same amount. As such, there is too much recombination, and not enough LD, in the simulations.

a. It could be that the N_e estimate is inferred from levels of genetic variation. However, where these populations were derived fairly recently from the wild, it is unlikely that genetic variation has reached equilibrium, so observed heterozygosity will not provide a good estimate of the strength of drift. N_e that is relevant for drift will be determined by census size and factors that reduce N_e (variation in reproductive success, selection, etc...)

(2) In arguing that LD is not likely to be responsible for the observed patterns, the authors note that signatures of selection decay over a span of 100bp from a selected site (line 202). Yet, the reference genome is "too fragmented to directly observe whether selected SNPs were clustered" (lines 630-631). These two statements are in tension with one another. It seems that there should be little power to say much about the decay in signatures of selection, given the fragmentation of the reference genome.

Second, I have concerns over the construction of background distributions for the population genomic inferences. The authors use Berg and Coop's Q_x statistic to infer polygenic adaptation, contrasting experiment-selected loci with background loci. The calculation of Q_x explicitly considers allele frequencies, and thus the background loci should be selected to have the same distribution of allele frequencies as the experiment-selected loci. There is no mention of this being done. However, the fact that the experiment-selected loci tend to show greater minor allele frequencies than the rest of the genome (lines 442-443) raises the possibility that Q_x differs because of allele frequency differences.

Because of these concerns, unfortunately I am not confident in the authors' conclusions.

Responses to Reviewers' Comments:

Reviewer comments are in black font. Author responses are in dark blue font.

Reviewer #1 (Remarks to the Author):

This paper by Stern et al. adds an exciting new approach to the previous work done in the *Eurytemora affinis* copepod species on adaptation to changing environmental conditions. The use of replicated selection lines combined with a survey of genomic variation in natural populations of this species in the Baltic region make this a valuable contribution. I think that the implications for evolutionary understanding and insights into how species can handle changing environmental conditions will make this interesting to a broader audience. Overall, I found the paper well-written with most of the analyses well-explained and appropriately interpreted. I do have a few concerns that could impact the framing of the broader message about the importance of epistasis that I think should be addressed and some other questions described below as well.

1. Polygenic model and epistasis: I think the findings for the models of polygenic loci that could explain patterns of parallelism are interesting but it is not clear to me that this clearly implicates epistatic interactions. My reading of this section is that they are adapting the polygenic model by just reducing the number of loci until they find the signature of parallel evolution that they find in their dataset. Are there other ways of doing modeling that more directly incorporate epistatic effects and how its existence would directly impact patterns of parallel evolution? In line 347 they mention that models are being developed that do incorporate epistasis but it doesn't seem like these are the models that they used. If the 30 gene model of polygenic inheritance fits that pattern of parallel evolution does that necessarily imply that widespread epistasis is the best explanation for why they see so many more SNPs under selection?

We completely agree with this reviewer's excellent suggestion. We had actually considered including epistatic model simulations in the original submission, but did not do so due to the length of the paper. In this revision, we now included several explicit models of epistasis in our simulations. We found that the high degree of parallel evolution observed in our data could be recreated by simulating epistatic effects among loci. We performed extensive simulations of polygenic adaptation in SLiM 3 (Haller 2019) under a variety of different models of epistasis (Supplementary Table 3). These models included several quantitative fitness functions implemented in MimicrEE2 (Vlachos and Kofler 2018), as well as an extension of Keightley and Otto's (2006) characterization of directional (positive or negative) epistasis. We found that simulating strong positive epistasis among selected loci could increase the Jaccard index (overlap) for selected alleles by 27% relative to a model without epistasis, and the degree of parallelism closely matched our real data.

We also compared the effects of epistasis relative to other potential factors influencing parallelism, such as the effects of selection from balanced polymorphisms and physical linkage (and recombination rate). We modelled the effect of balanced polymorphism by simulating starting allele frequencies from mutation-drift balance (neutral) and from their empirical frequencies (potentially balanced). We did find that starting allele frequency can increase the degree of parallelism, as we have found previously (Stern & Lee, 2020), but only by ~5% (versus 27% for epistasis). When we simulated selection with physical

linkage under different recombination rates, we found that physical linkage slightly decreases parallelism by approximately 0.5% in simulated data.

Thus, our model simulations of epistasis reinforce our conclusion that positive synergistic epistasis could promote parallelism on a polygenic scale and produce the high degree of parallelism observed in our experimental data.

2. Could a long history of balanced polymorphism across a large number of loci without epistasis explain the observed patterns (maybe via increased linkage disequilibrium by tending to be in the same individuals due to a history of environmental selection but not via physical linkage)?

We agree that this phenomenon could create a lot of linkage disequilibrium (LD). However, salinity is not a binomial environmental factor, but generally changes gradually across space and time, with animals with overlapping generations going in and out of diapause across seasons as habitat salinity changes. Therefore, recombination is likely to occur between individuals from different salinity environments and likely to break up the LD due to such an effect.

3. Detection of parallelism—it isn't clear to me exactly how the selected SNPs are determined; can a SNP be found significant for selection in only one line of the experiment or does it depend on repeated selection across the lines. If it is the latter this would seem to bias it towards detecting the cases of parallel selection.

Thank you for raising this point, which warrants further clarification in our text. Selected SNPs were determined using three methods: (1) Chi-square tests to detect significant frequency changes in individual treatment lines, (2) CMH tests to detect significant frequency changes using all treatment lines, and (3) Linear mixed-models to detect significantly different frequency trajectories between all treatment and control lines. All of these tests accounted for variance due to Pool-seq sampling. The intersection of all these significant SNPs were input into the selected haplotype block reconstruction method (i.e. *haplovalidate*), and therefore selected alleles could include both those with line-specific and shared signatures of selection. We note that a similar procedure of including both line-specific and shared SNPs in haplotype-block estimation has been used in a previous study that uncovered considerable heterogeneity (i.e. non-parallelism) among selection lines (Barghi et al. 2019).

In response to this comment, we added the following sentences to clarify our methods on lines 170-177:

“To detect single-nucleotide polymorphisms (SNPs) with signatures of selection in response to salinity decline, we performed whole-genome pooled sequencing at generations zero, six, and ten. We used Cochran–Mantel–Haenszel (CMH) tests and Chi-square tests to detect SNPs with frequency shifts that were greater than expected, relative to a model with only genetic drift, both across replicate lines and within individual replicate lines (Fig. 1a, upper panel). We also used linear mixed models (LMMs) to detect SNPs with frequency trajectories that differed significantly between treatment and control lines (Fig. 1a, lower panel; see Methods). In total,

these tests uncovered 18,072 candidate SNPs with signatures of selection (out of a total of 353,188 SNPs tested).”

4. Physical linkage and inversions—If the genome assembly is rather fragmented and overall patterns of physical linkage are harder to determine, it seems like this weakens some of the arguments to a degree. For the analysis of lack of mapping of one read of a paired end read, have other studies shown this to be an effective way of looking for inversions in this situation? It seems plausible, but there could also be a lot of other artifacts and other genome assembly issues that could make it less likely to work.

In this revision, we have now taken advantage of a new highly contiguous genome assembly for the study species based on PacBio sequencing and Hi-C. Using this genome, we were able to arrange and orient contigs along four large scaffolds (putative chromosomes). We now have a better understanding of how selected loci are distributed across the genome and have now presented a Manhattan plot of SNP signatures of selection on one scaffold in Figure 1a. Furthermore, as stated above, we have now used a method to identify selected haplotype blocks to group linked SNPs together and explicitly account for the effect of linkage on the number of selected loci and degree of parallelism.

5. Line 146 The statement ‘help resolve’ seems a bit strong—maybe something like contribute to our understanding? Similarly in Line 154, the ‘with great precision’ feels a bit strong as well.

We agree with these comments and have changed ‘help resolved’ to ‘provide unique insights into’ (Line 144) and removed the phrase “with great precision”.

6. Figure 1a: how do you get the allele frequencies averaging ~0.5 when presumably natural variation was of all frequencies at the start? Maybe it is just that the axis label is confusing as it sounds like you did a regression to get these values (and it is not raw changes in allele frequencies plotted).

We agree that this figure was a bit confusing and have created a new figure (Fig. 1b) that displays actual allele frequency changes in a clearer manner. In this new figure, the average frequencies of all individual 121 selected haplotype blocks (alleles) are displayed for the treatment lines in each generation (grey lines). We also show the average selected allele frequency across the treatment lines (purple lines) and control lines (yellow line), along with an expected distribution from neutral simulations (blue shaded area).

Reviewer #2 (Remarks to the Author):

In manuscript ‘Parallel polygenic adaptation driven by epistasis in laboratory and wild populations of a Baltic Sea copepod’ authors have evolved copepods in low salinity environment and investigated the genomic signatures of selection. They have performed computer simulations mimicking selective sweep and polygenic adaptation of quantitative traits and concluded that synergistic epistasis is prevalent in their study. Although authors emphasize that epistasis is the reason the empirical data doesn’t fit the simulated evolutionary scenarios, epistasis is not explicitly modeled and simulated; the lack of fit of empirical data to the simulated ones has been interpreted as epistasis. The authors are correct that the role of epistasis in adaptation is not well studied, but modeling epistasis is very difficult as the information about the functional

network of genes in many sexual eukaryotes is still missing. But I'm afraid that the analysis and simulations of the manuscript doesn't fully support the role of epistasis in adaptation in this study system. There are also some aspects of data analysis and simulations that need further investigation.

Major comments:

1. There is a general mis-understanding/presentation of polygenic adaptation (e.g. L139); That, polygenic adaptation would not result in parallel response. Many factors including distance to the new trait optimum, heterogeneity of alleles (effect size and starting frequencies), epistasis, pleiotropy, and GxE affect polygenic adaptation, and in conditions genomic response can be parallel under polygenic adaptation. If the new trait optimum is far from the starting phenotype of population or few adaptive alleles are present or effect size of alleles are large, the genomic response of polygenic adaptation in fact will be parallel and quite similar to selective sweep. The tone of discussion in this manuscript, however, is that all previous theoretical or empirical studies which have shown or suggested non-parallel response across replicate populations have missed the point, or overlooked the true signal due to the noise in dataset. Many aspects of data analysis and simulations should be done more thoroughly to make sure that the observed pattern is true.

We agree that our presentation of polygenic adaptation was not sufficiently comprehensive. We have now included in the Introduction (Lines 70-83) a discussion of additional scenarios, both theoretical and empirical, in which polygenic adaptation could be parallel.

In addition, our simulations now account for the many of the factors stated above (distance to the trait optimum, heterogeneous effect sizes), or investigate their effects (starting frequency, linkage) on the degree of parallelism.

Regarding distance to the trait optimum: During our simulations under quantitative fitness function, the trait optimum or plateau was chosen such that the distribution of population phenotypes had just reached the trait optimum or plateau by generation 10 (an example of initial and final population phenotype distributions is shown by the horizontal bars in Supplementary Fig. 1). Therefore, populations did not spend time in a 'non-adaptive' phase and our expected parallelism under this model is not reduced. Despite this fact, the levels of parallelism observed under these quantitative fitness functions did not approach the observed levels of parallelism.

Regarding the number of adaptive alleles: Through simulation, we varied the number of alleles contributing to fitness under all of our models of fitness. While we did find levels of parallelism similar to the observed when the number of loci is small (Fig. 2b, Supplementary Figs. 4,5), only strong positive epistasis was able to replicate the observed high level of parallelism when all 121 alleles (haplotype blocks) were included.

Regarding the effect sizes: While the large effect sizes inferred during our analysis (Fig. 1d) certainly did contribute to the levels of parallelism observed, this effect on parallelism was accounted for in our simulations under the multiplicative fitness model. We found that selection acting on individual alleles (i.e. the multiplicative model) with our high observed effect sizes only generated pairwise Jaccard Indices of 0.520 in generation ten, versus 0.795 for our empirical data (Figs. 2a, 2b). Thus, a factor beyond large effect sizes must account for the observed degree of parallelism.

Regarding starting frequencies: We began several analyses under putatively neutral allele frequencies (sampled from SNPs that did not return any signature of selection), which tended to be lower than the starting frequencies of the selected alleles. Furthermore, we performed another round of simulation where all selected alleles started at a frequency of 0.5, when selection will be most efficient. We found that starting frequency impacted the expected level of parallelism by a low amount, of up to 0.05 in the Jaccard index. Moreover, this factor did not appear to impact levels of parallelism as much as the shape of the fitness function (Supplementary Fig. 3).

Overall, even when the above stated effects were explicitly modeled, our observed degree of parallelism could not be re-created without including strong epistatic effects among loci.

Moreover, the authors don't show or even model epistasis in their simulation, they show that the observed parallelism in the empirical data fits a model with fewer alleles of larger effect size. The parsimonious explanation/interpretation could be that the number of adaptive alleles in this study has been overestimated. If authors believe that their findings support prevalence of epistasis in polygenic adaptation, epistasis should be modeled explicitly. In line 89-92, it is mentioned that the loci with epistasis are functionally linked and their interactions are not additive. While in the simulations, effects are additive and no functional linkage (or even physical linkage) is considered.

As mentioned above, this is an excellent suggestion that we considered in the previous submission. In this revision, we now explicitly modelled epistasis using a variety of both multiplicative and quantitative fitness functions. We found that strong positive epistasis greatly increased the degree of parallelism in the simulated data and fit our data better than including strong physical linkage (Fig. 2, Supplementary Fig. 2). Regarding the over-estimation of adaptive alleles, we now took into account linkage among SNPs using the haplotype reconstruction method recommended below to significantly reduce the inferred number of selected loci and visualized the distribution of selected loci across the genome (e.g. Fig. 1a). Nevertheless, we found that the observed degree of parallelism greatly exceeded the expectations for the 121 selected haplotype blocks (Fig. 2). These additional analyses made our inferences far more rigorous and conclusions more robust.

2. Authors state that parallelism and synergistic epistasis in this study are consistent with a complex physiological trait (L45-47), but in Discussion and GO term analysis section (L400-402) where the authors claim to have identified the true functional/selected trait, it is mentioned that the studied trait (adaptation to low salinity) is not as complex as other traits such as adaptation to temperature and it's simpler. So if the studied trait is not as complex as temperature adaptation, then the trait might be oligogenic with few contributing loci, and the true number of selected loci might be less.

Environmental temperature is particularly extreme in inducing a global physiological response, affecting all metabolic processes in an organism, especially in ectotherms. Nearly every physiological trait will likely be less polygenic than temperature adaptation, while still being encoded by many genes. Our salinity adaptation trait is likely encoded by hundreds of genes, given the number of cooperating ion transporters and other osmoregulatory genes. Indeed, in our new analysis in which we grouped selected SNPs in

haplotype blocks, the number of selected alleles was lower than our original estimation (121 haplotype blocks vs. 1156 SNPs). However, we still found that the degree of parallelism for these 121 loci was considerably higher than expected based on models of positive and directional selection, both with and without linkage (Supplementary Fig. 2).

3. In Fig. 2, the authors show that the parallelism in their study is even higher than sweep, while sweep is conditional on fixation of all alleles, I am curious to see what is the interpretation that the experiment has more parallelism than selective sweep model would predict? If the parallelism in empirical data is similar to simulations of 30 loci, how can the authors reject the hypothesis that the true number of selected targets is less than the estimated number of selected SNPs (1156)? Considering that linkage has not been taken into account in this experiment which has small population size and probably high drift, and selection should be estimated using a more suitable software, and both number of loci and selection coefficient are important parameters of simulations, further investigation of these factors is needed (see below).

The reviewer makes excellent points here. Our simulations were actually not conditional on allelic fixation. Therefore, we no longer refer to this model as the “selective sweep” model, but instead call this model “multiplicative fitness”, which we believe describes the model more accurately (Supplementary Table 3). As discussed above, we now explicitly consider linkage in our determination of selected loci. While it is possible that we have underestimated N_e and the selection coefficients, we have now taken multiple approaches toward estimating both N_e (Point #8; Supplementary Table 9; Lines 531-538) and the selection coefficients (Point #5; Lines 585-592) to increase confidence in these important parameter estimates.

Data analysis:

4. L617: The simulations that are used to estimate the number of loci varies both the number of loci and their effect sizes. In fact, the results show that simulations with fewer loci and different effect size fits their empirical data better. Authors should perform additional analysis (looking for selected haplotype blocks rather than assuming independence among selected SNPs) to account for linkage among the selected SNPs (which might reduce the number of selected sites) and use other tools to estimate selection coefficient (which affects the trajectory of alleles). These 2 parameters determine the allele trajectory and level of parallelism among simulated replicates, which heavily affects the conclusions of this paper (see below). Are the selected SNPs localized in specific regions of genome? Give the small population size and high effect of drift, could the selected SNPs be linked and not independent? If plotted in a Manhattan plot, do the selected SNPs form peaks (linked selection) or are they scattered across the genome? There are many tools developed to reconstruct the partial selected haplotype block using time series data. I suggest authors use one of the tools to account for the effect of linkage in the number of selected SNPs.

We very much appreciate these detailed and useful suggestions. We have now performed additional analyses to increase confidence in the number of selected loci and selection coefficients (see Point #5 below for discussion of selection coefficients). Taking advantage of a new highly contiguous genome assembly for our study species complex (based on PacBio and Hi-C sequencing, see Point 4, Reviewer #1 above), we were able to visualize selected SNPs in a Manhattan plot and observed that they are scattered across the genome (e.g. Fig. 1a for a visualization across one scaffold). As suggested, we have now grouped significant SNPs into putatively independent selected haplotype blocks using *haplovalidate* (Otte & Schlötterer 2021). This process has reduced our inferred

number of selected loci from 1156 to 121. Nevertheless, we found that the observed degree of parallelism for the 121 loci was exceptionally high (see Fig. 2). The high degree of parallelism was much greater than expected using simulations under the population genetic “multiplicative fitness” model using our real selection coefficients, population sizes, and linkage structure (See section *Exceptional genomic parallelism across laboratory lines*).

5. L581: There are many tools developed specifically for estimation of selection coefficient. A recent benchmarking study (<https://link.springer.com/article/10.1186/s13059-019-1770-8>) has shown that CLEAR provide the most accurate estimation of s . I suggest authors use one of these established methods and compare their estimated s .

As suggested, we used two additional methods to estimate s , namely CLEAR and Nest (now part of the *poolSeq* R package). We ultimately decided to use the estimates from our original method for several reasons: (1) Our method (using linear mixed models to regress transformed allele frequencies against generation) is nearly identical to Nest, but also takes into account variance due to Pool-seq sampling and the random effect of replicate line, as recommended by the authors of Nest, (2) Our estimates of s are highly correlated with those estimated with Nest ($R=0.79$, $df=119$, $P \ll 0.01$), and (3) We found that CLEAR was less precise in its estimates of s , which are only reported by the software in increments of 0.05 and are therefore not useful for our study. The study linked above, where CLEAR performed well, simulated an estimated s of 0 and 0.05 to evaluate CLEAR’s performance on simulated data.

6. The CMH test identifies consistent allele frequency changes across majority of replicates, so if an allele is selected in few replicates only, it would have high p-value in the CMH test. Previous studies that have been cited in this manuscript have used Chi-square or Fisher’s exact test to identify replicate-specific selection (Barghi et al. 2019, Otte et al. 2020). It would be good if authors perform additional analysis to identify replicate-specific response, if present.

Thank you, this is a valid point. We have now incorporated Chi-square tests to identify SNPs with signatures of replicate-specific selection (Line 173). The SNPs with signatures of replicate-specific selection were also used to identify selected haplotype blocks, and therefore our inferred selected alleles are now not biased toward the detection of shared frequency changes.

7. The authors have used linear regression to contrast selected and control lines. If CMH test is used to contrast control and selected lines, will the same SNPs be identified as outliers?

Unfortunately, our sampling design, with fewer control than treatment lines, did not lend itself to contrasting control and selected lines with a CMH test. Furthermore, a CMH test contrasting control and selected lines would detect SNPs with different *frequencies* at a given timepoint, while we were interested in SNPs with different *frequency trajectories* (i.e., divergence from the ancestral population).

Simulations:

8. The population size seems to be 500, how can the estimated N_e be 2000, 4 times higher than the

census population size? This parameter plays a very important role in simulations, as 2000 is a large N_e and will result in low drift and high parallelism. PoolSeq package is used for N_e estimation, it would be good if authors specify which functions/parameters were used (Couldn't find any relevant command in the github)? N_e is normally computed in windows along the genome, is the reported N_e the median of all windows? Are low frequency SNPs filtered, since these might inflate the estimated N_e ? The authors perform simulations mimicking experimental design of Tobler et al. (2014) and their study to show that their results are not biased by linkage and the number of identified selected SNPs are not inflated by long-range LD. N_e play an important role in these simulations. Authors should clarify the estimated N_e (see below).

This is an important point and we have now performed additional estimates of N_e . We performed a more thorough and systematic estimation of the effective population size by exploring different methods and parameter settings in the estimates (Lines 531-538, Supplementary Table 9). Our initial census size estimate of 500 individuals was a rough estimate that was most certainly a gross underestimate, given that we did not count the fertilized eggs and larvae in each replicate. We therefore decided to rely on the genomic data to estimate N_e . We have now computed N_e using the procedure recommended by the reviewer (computed in windows and taking the median). We explored methods with and without incorporating census size, and we filtered SNPs with a $MAF < 0.05$. The various estimates are now presented in Supplementary Table 9 and appear to converge on an average estimate of ~ 1750 across lines. These new estimates were incorporated into all analyses and simulations that rely on effective population sizes.

9. The authors have looked at decay of LD (L202) and concluded that their study doesn't suffer from the effect of linked selection/linkage. But the decay of LD in Tobler et al. study is quite similar. A follow-up study on Tobler study on the same dataset has shown that despite the quick decay of LD at 200bp, long-range LD can still be prevalent, not due to segregating inversions but frequency increase of haplotype blocks (Franssen et al. 2014). I suggest in order to have a more accurate estimation of selected sites (which are quite important for the simulations) effects of linkage should be taken into account.

As noted above (Point #4, Reviewer #2), we have now explicitly accounted for linkage among selected SNPs using the method recommended by the reviewer, namely identifying selected haplotype blocks with *haplovalidate*. Therefore, we have removed from the paper this analysis and discussion of the decay of LD.

10. Authors have used MimicrEE2 for simulations to assess the effect of linkage but for simulations of evolutionary scenarios they have used other tools that don't account for linkage. Authors should account for linkage for their sweep (and polygenic adaptation) simulations. I suggest using MimicrEE2/SLiM or any other simulation tools for all simulations specifically to account for linkage among selected loci.

Following this excellent recommendation, we explicitly modelled the effect of linkage among selected loci in our simulations using SLiM 3 (Lines 267-276). Furthermore, we explored the effect of recombination rate on the degree of parallelism. These simulations did show an effect of linkage on parallelism, but this factor could only account for an average difference of 0.005 Jaccard index across models. This amount cannot account for the high degree of parallelism observed in our study (Supplementary Fig. 2).

11. L522-525: The simulated polygenic adaptation is a 'quantitative trait after a shift in the trait optimum'.

However, the selection regime in this study doesn't fit this description as the salinity of environment (the selective factor) has changed from generation 2-6. It is not obvious from the description of experiment whether only the surviving individuals were transferred to the next generation (truncating selection) or not (moving trait optimum). But the selection mode in the simulations of polygenic adaptation should be modified to fit the experiment.

In the experiment, we did not select or transfer individuals to the next generation. We ultimately decided to simulate under multiple different quantitative fitness functions (Supplementary Table 3) for two reasons: (1) Our experiment allowed individuals to survive and replicate naturally (i.e. we did not select and transfer individuals to the next generation), and therefore the exact mode of selection is unknown, and (2) Our experimental design included a moving optimum (salinity declined over generations 2-6), which was not perfectly captured by available fitness functions. Therefore, simulating under a range of different quantitative fitness functions could help to capture allele frequency dynamics under the quantitative genetic paradigm regardless of the exact (unknown) mode of selection.

12. For simulations of polygenic adaptation why 0.9 is chosen as the new optimum? how far is this new optimum from the phenotype of starting population in the simulations? Would authors observe similar results if they change the distance to trait optimum?

These are excellent points that we should consider. In this revised manuscript, the parameters of the fitness functions were chosen such that the distribution of population phenotypes had just reached the trait optimum by generation 10. Initial and final population phenotype distributions are shown by the horizontal bars in Supplementary Figure 1. While changing the distance to the trait optimum would impact the degree of parallelism, the quantitative fitness functions were constructed based on the starting phenotypes to impose very strong selection pressure to maximize the possible parallel evolution. Nevertheless, we find that none of the quantitative trait models could recreate the high levels of parallelism observed in our study (average Jaccard index across quantitative models at generation ten = 0.38 ± 0.03 SD).

13. 10 iterations are too few, at least 500-1000 simulations should be performed.

We have now performed 1000 simulations under each model and parameter combination, except for the analyses with variable numbers of loci, for which we performed 100 simulations each. Box plots displaying the distribution of Jaccard index values across the 1000 simulations are now displayed in Figure 2a, Supplementary Figure 2 and Supplementary Figure 3.

14. The estimated Jaccard index for simulations is pooled for all 10 iterations. To properly mimic the experimental design, it should be computed for each set of 10 replicates, and the mean and SD across total iterations be reported.

We have now calculated the mean Jaccard index between replicates for each simulation and used those values to summarize the mean and SD across all simulation iterations.

15. If the distance to trait optimum is the same in polygenic adaptation simulations for 1156 and 30 loci, and in each simulation loci have different effect sizes (i.e. each of 30 loci have higher effect size than 1156 loci), it might take much longer for the population with 1156 loci to reach the trait optimum, while in the simulation with 30 loci the population might have reached the new trait optimum after 10 generations. In addition to trajectories of allele frequencies, the phenotype of populations should also be checked to make sure populations in both simulations have reached the trait optimum, otherwise it's like comparison of apple and oranges.

Thank you for raising this valid point. In this revision, we now adjust the fitness function so that the distance to the trait optimum is equal among simulations, regardless of the mean starting phenotype (Supplementary Table 3; Supplementary Methods, Section 5). To accomplish this, we introduce a “horizontal shift” parameter into the simulations defined to be the difference between the mean initial phenotype in the current simulation and the mean initial phenotype in the simulations of 121 alleles with empirical parameters (Supplementary Table 3). This modification allows for a more appropriate comparison between simulations with different numbers of loci. Furthermore, we have now included in the text the point that a small number of large effect loci will increase the rate of adaptation under the quantitative genetic model (Lines 336-339).

16. L267-269 (Supplementary figure 4) why 0.1 frequency change is used as threshold for selected alleles? the expected frequency change depends on the starting frequency of alleles. This threshold should be adapted depending on the starting frequency of alleles.

In response to this comment, we now use a cutoff based on the top 0.1% of 10,000 neutral simulations, accounting for the starting frequency (Lines 579-584).

17. The details of parameters used for the simulations should be explicitly specified not just described.

The simulation parameters are now presented explicitly in Supplementary Table 4.

Minor comments:

18. L73-75: There are many natural systems, for example sticklebacks, that show highly parallel response, mostly involving large effect alleles.

Our original paper did cite several review papers that discussed the threespine stickleback (Line 60) in reference to the observation of parallel evolution for individual large-effect alleles. In this revision, we have now included additional discussion of situations in which polygenic adaptation has been shown to be somewhat parallel (Lines 75-83 in the Introduction):

“While some genomic studies have found more genetic parallelism than expected by chance (e.g. ^{8,16,19-21}), such studies tended to observe fewer than 50% of selected alleles in common between populations⁷. Several factors have been proposed that could promote high levels of parallelism, including large distance to the new trait optimum, low divergence between populations, higher parallelism for large effect and high

frequency alleles, pleiotropy, and others^{6,22}. Yet, the relative impacts of these factors in promoting molecular parallelism remain unknown.”

19. L132: It would be helpful if authors elaborate how references 35 and 36 suggest the prevalence of epistasis in this study system.

We have removed this sentence for clarity.

20. L144-146: Parallel polygenic adaptation has been repeatedly observed in many organisms, in natural and experimental observations. The most famous experimental populations is study of longevity in *Drosophila melanogaster* evolved over 800 generations with many replicates (<https://academic.oup.com/mbe/article/34/4/831/2897202>).

Thank you for this point. We have now included a more nuanced discussion of situations in which parallel polygenic adaptation has been observed and to what degree (See point #18, Reviewer #2) and also cited the study mentioned in this comment. The primary contribution of our study is in providing potential mechanisms of parallel polygenic evolution.

21. Is there any phenotypic information about these selected replicates? Have they converged phenotypically?

We did not take phenotypic measurements throughout the course of this experiment, as doing so was beyond the scope of this particular study. However, previous experiments using *E. affinis* complex populations have demonstrated phenotypic evolution and convergence after selection in fresh water following several generations of laboratory evolution, such as convergence of physiological tolerance, life history traits, and ion transporter activity and expression (Lee et al. 2011; Lee et al. 2007).

22. L527: The diet of selected populations has changed in the course of selection experiment. What is the effect of diet on the genomic response? It would be good to clarify why the diet has been changed for readers who are not familiar with this study system.

The use of *Rhodomonas* algae (or some other Cryptophyte) as a food source for *E. affinis* is key to their survival in the laboratory (Lee et al. 2013). Individual *Rhodomonas* species are highly sensitive to salinity, such that the same algal species cannot survive in both salt and fresh water. Therefore, changing the diet throughout the course of the experiment was an unavoidable factor than could not be decoupled from the salinity environment. Although, throughout the experiment we used closely related congeners that differ in salinity tolerance, but are both rich in long chain polyunsaturated fatty acids. This point has been made clear in the text (lines 503-508). Our prior experiments have shown that both saline and freshwater copepod populations survive well on both algal species at the intermediate salinity of 5 PSU.

23. L558: what is the correlation of SNP frequencies between the two starting population samples?

We found the SNP frequencies between the two starting samples to be highly correlated ($R^2 = 0.90$).

24. Which time point was used for the CMH test, generation 6 or 10?

We used a version of the CMH test that can use multiple timepoints (i.e. both generations 6 and 10) to better account for genetic drift (Spitzer et al. 2020) and made this point clear in the text (Lines 542-545).

25. The authors compare the similarity of replicates using Jaccard index but they often refer to it as 'pairwise overlap', I suggest they use the conventional term 'Jaccard index' and cite the original paper.

We have now used the term "Jaccard index" throughout the manuscript and cited the original paper at first use.

26. L646: DAVID doesn't account for gene length which affects the enrichment analysis, it would be goof to use a software that takes gene length into account.

We have now used the software GoWinda, which does take into account gene length and SNP density. As our set of selected SNPs has now changed, we have also modified our section about enriched GO terms (Lines 197-206). In this version of the manuscript, we used all of the SNPs characterizing a selected haplotype block as input to GoWinda, following Barghi et al. (2019). After these changes, we still found GO terms related to ion transport and osmoregulation to be the most overrepresented (Lines 202-206; Supplementary Table 2).

27. Fig. 1a: are the trajectories of the same SNPs shown for both selected and control lines? if yes, why the starting frequencies differ among these lines?

In light of this comment, we have modified this figure (now Figure 1b) to better display the selected allele frequency trajectories. The average frequencies of all 121 selected alleles are displayed for the treatment lines in each generation (grey lines). We also show the average selected allele frequency across the treatment lines (purple lines) and control lines (yellow line), along with an expected distribution from neutral simulations (blue shaded area).

28. Fig. 1d. I don't think s should be computed/reported for non-selected SNPs!

Thank you for this valid point. We now only report selection coefficients for selected alleles.

29. Fig. 2. The summary statistics in Fig. 2c is Replicate frequency spectrum, it would be good to use the proper name and cite the original paper.

We now used the term Replicate Frequency Spectrum (line 225) and cite the original paper (Barghi et al. 2019) at first use.

30. L181: how is the frequency change of 0.1 consistent with polygenic adaptation, the reference is not related to this threshold.

We agree that this section is a bit confusing and that the reference did not exactly support this statement. We have decided to remove this sentence and instead display the frequency shifts in Figure 1b and selection coefficients in Figure 1c. We also report the average frequency shifts (e.g. “9.9% at generation six and 12.8% at generation ten”).

31. L364-366: what is the direct evidence that the effect of epistasis is more prevalent than starting frequency of allele?

In this revision, our explicit modelling of epistasis in our simulations enabled us to quantify the relative effects of epistasis and high frequency alleles. By comparing the average Jaccard Index with simulations of our empirical starting frequencies, neutral starting frequencies, and all alleles starting at 0.5, we found that starting frequency could impact the degree of parallelism by approximately 0.05 in Jaccard index on average across models (Lines 368-376; Supplementary Fig. 3). In contrast, we found that the inclusion of positive epistatic effects among loci (“Positive epistasis” model in Supplementary Table 3) increased the Jaccard index in simulations from 0.52 to 0.79 (Fig. 2a). This increase of 0.27 in Jaccard index due to positive epistasis is considerably larger than the 0.05 due to starting frequency.

32. L389-391: are these GO terms significantly enriched?

Yes, the GO terms in the text are significantly enriched. Results of the statistical tests for enrichment are now in Supplementary Table 2.

33. Fig. 3c and d. Is the MAF folded? if not, how the derived allele was determined?

Yes, minor allele frequencies (MAFs) are folded by definition. This point is now stated in the figure legend.

Reviewer #3 (Remarks to the Author):

In this study, the authors investigate adaptation of the copepod *Eurytemora affinis* to low salinity environments, using both laboratory selection experiments and population genomic studies of natural populations. They report evidence for highly polygenic adaptation, with dozens to hundreds of loci under selection. Parallelism is much more frequent than expected under standard models of polygenic adaptation; the authors argue that widespread positive epistasis could explain this finding.

The experimental design and dataset are very impressive. However, I am concerned about several aspects of the analysis, and thus with the resulting conclusions

1. First, I am concerned about the authors' treatment of linkage disequilibrium (LD). LD could in principle account for several notable aspects of the data, including the seemingly large number of loci under selection (since some of these loci could simply be hitchhiking with selected sites), and the high level of parallelism (since linked sites would be expected to show parallel trajectories). The authors are well aware of these issues, and undertake extensive simulations in an attempt to address them. A few concerns here: (1) In the simulations, the authors use an N_e of 2000 (lines 195; 602-603; 611). I don't see this number justified anywhere - where does it come from? 2000 strikes me as very high – census size is ~500, and N_e is typically less than census size, often by a lot. In this case, the authors will have overestimated N_e by an order of magnitude or more, and the population recombination rate for the simulations will be off by the same amount. As such, there is too much recombination, and not enough LD, in the simulations.

a. It could be that the N_e estimate is inferred from levels of genetic variation. However, where these populations were derived fairly recently from the wild, it is unlikely that genetic variation has reached equilibrium, so observed heterozygosity will not provide a good estimate of the strength of drift. N_e that is relevant for drift will be determined by census size and factors that reduce N_e (variation in reproductive success, selection, etc...)

Thank you for your positive comments and for bringing these points to our attention. To address the first concern (#1, above), we now provide a more systematic estimation of N_e (Supplementary Table 9) that converges on an average estimate of ~1750. As noted above in this document (Point #8 under reviewer #2), we likely underestimated the census size considerably, as fertilized eggs and larvae are difficult to count visually. The estimate of N_e is based on the variance in SNP frequencies across our sampled timepoints and is therefore an accurate method of N_e estimation, as has been shown in other studies (e.g. Jónás et al. 2016, Taus et al. 2017, Barghi et al. 2019). While this estimate of N_e is relatively high, estimates for wild populations of this species complex using observed heterozygosity are orders of magnitude higher (Winkler, Dodson, & Lee 2008). Given that our estimates of N_e are taken directly from the variance in allele frequencies over time, they should accurately represent the strength of genetic drift during the experiment.

2. (2) In arguing that LD is not likely to be responsible for the observed patterns, the authors note that signatures of selection decay over a span of 100bp from a selected site (line 202). Yet, the reference genome is “too fragmented to directly observe whether selected SNPs were clustered” (lines 630-631). These two statements are in tension with one another. It seems that there should be little power to say much about the decay in signatures of selection, given the fragmentation of the reference genome.

In response to this important point, we have now taken advantage of a new highly contiguous genome assembly for the study species complex based on PacBio and Hi-C sequencing. Using this new genome, we were able to arrange and orient contigs along four large scaffolds (putative chromosomes). We now have a better understanding of how selected loci are distributed across the genome and have now presented a Manhattan plot of SNP signatures of selection on one scaffold in Figure 1a. Furthermore, we have now used a method to group linked significant SNPs together into 121 putatively independent selected haplotype-blocks. These improvements allow us to explicitly account for the effect of linkage on the number of selected loci and degree of parallelism. Our results now describe patterns of frequency shifts and parallel evolution for these 121 alleles. We find very high levels of parallelism for these haplotype-blocks with ~79% of alleles

experiencing shared allele frequency shifts between replicate lines at generation ten, much higher than expected using baseline simulations (Fig. 2a).

Furthermore, we also now evaluate the effect of linkage and recombination by including these factors as part of our simulations using both realistic and decreased recombination rates (Supplementary Fig. 2). Even after accounting for physical linkage and taking into account other factors, such as starting frequency, effect size, and distance to the trait optimum, we still find that positive epistasis is the best explanation for the observed degree of parallelism. Physical linkage appears to only have a small impact on the degree of parallelism (Supplementary Figure 2), as has been observed in other studies (e.g. Barghi et al. 2019).

3. Second, I have concerns over the construction of background distributions for the population genomic inferences. The authors use Berg and Coop's Q_x statistic to infer polygenic adaptation, contrasting experiment-selected loci with background loci. The calculation of Q_x explicitly considers allele frequencies, and thus the background loci should be selected to have the same distribution of allele frequencies as the experiment-selected loci. There is no mention of this being done. However, the fact that the experiment-selected loci tend to show greater minor allele frequencies than the rest of the genome (lines 442-443) raises the possibility that Q_x differs because of allele frequency differences. Because of these concerns, unfortunately I am not confident in the authors' conclusions.

This is an important point, and we have now made it clear in the text that the background distribution was, in fact, constructed by sampling SNPs that were matched in minor allele frequency with the focal SNPs (Lines 656-657). Therefore, the observed high Q_x would not have been the result of a different MAF distribution.

Reviewers' Comments:

Reviewer #1:

Remarks to the Author:

I think that the authors have done a good job of addressing my concerns from my previous review and am supportive of having this published with minor tweaks. I like their incorporation of explicit modeling of epistasis showing that the results are consistent with these models and less consistent with a broad set of other models. In general, I find their discussion of the results in line with this framing (i.e. positive epistasis can explain the results but has not directly been demonstrated). The one place where I might suggest a change would be the title. Perhaps a more conditional phrase for the role of epistasis should be used, something like 'may be promoted by epistasis'. The addition of new results with chromosome-scale linkage is also very helpful.

I have a few other minor comments as well:

line 41-confusing sentence structure and not clear what 'evolutionary response architecture' is.

line 132 is confusing and may be missing some punctuation

line 156 Fig. 1: Why was this scaffold selected (is it representative or the one with the largest effects)?

Data availability-will the pseudo-reference genome for the European populations be made available in some fashion? Probably would be in some format other than GenBank or I5K

Line 245: It would be good if they at least briefly explain how the results in Fig 2c are obtained and what the terms mean. I may have missed it but didn't find an obvious explanation in methods or experimental methods.

Reviewer #2:

Remarks to the Author:

Authors have made great efforts to incorporate the comments and suggestions of all reviewers. The analysis for the identification of selected alleles has improved and more simulations have been performed. I also very much appreciated embedding figures in the text which made reviewing easier. The role of positive epistasis in the observed high parallelism among replicates relies on the performed simulations and computed summary statistic: Jaccard index. However, I found the description of simulations a bit confusing. I would appreciate it if details of simulations are clarified.

1. Given the high starting frequency of alleles and strong selection coefficient of the selected alleles, overlap among replicates might also be high under neutrality. It would be interesting to see how the Jaccard index under neutrality is different from the empirical data.

2. L629: How is Jaccard index computed for simulated data? Which alleles are considered selected/responding in a replicate in simulations?

3. In Supplementary Fig. 3 why is the empirical Jaccard index lower for SNPs than for blocks? In previous version of the paper with 1156 SNPs, the Jaccard index at generation 10 was around 0.9.

4. I am still surprised that a population set up from 50 individuals with not much recombination can have an estimated N_e of 1750. There should be many uncounted fertilized eggs and larvae for this to happen. It seems that N_e is estimated using one method, it would be good to use another method just to make sure more or less similar estimated are obtained.

5. Supplementary data, section 5: Simulating alleles with starting frequency of 0.5 seems unrealistic even for experimental populations. The effect of starting frequency on parallelism can also be tested by checking whether the high frequency alleles are over-represented as the selected alleles identified in several replicates in simulations using empirical starting allele frequencies.

6. Supplementary data, section 5: 'Therefore, it is possible that if, for instance, 100 loci contributed to adaptation, all 100 could be located on the same haplotype block or they could all be located on their own haplotype block and be unlinked.' Would you please clarify whether this means simulations were performed with linkage (or without linkage)?
7. Parallelism of empirical and positive epistasis model match perfectly in Fig. 2b but not in Fig. S2 in simulations with and without linkage. What is the difference between the simulation parameters of these two figures in terms of number of alleles, starting frequency and s (I assume based on the text that all parameters should be similar to the empirical data)?
8. What is the difference between selective sweep/directional selection and multiplicative model?
9. In Supplementary Fig. 3 why there is such subtle difference between simulations of empirical starting frequencies and those with starting frequency of 0.5? Is linkage considered in these simulations?
10. The details of simulations which the conclusion of paper heavily relies on are mostly in supplementary files, if there is no page limit from the journal, it would be good to include this important information in the main text.
11. The argument about the synergistic effect of ion transporters is interesting and fits well with the positive epistasis notion. However, the GO analysis has been performed using all the SNPs found in the haplotype; this assumes that all these SNPs are true positives. But the motivation for clustering SNPs into blocks is linked selection. The list of ion transporter genes in Table 1 also shows that some of these genes are present on the same blocks. How does the possibility that not all the genes in the selected haplotype blocks might be selected affect the argument about the synergistic effect of ion transporter genes?
12. 339-345: These arguments are based on the assumption that 121 identified haplotype blocks are independent. Just by eyeballing, several of the haplotype blocks (yellow, beige or brownish in the middle of the scaffold) in Fig. 1a seem very close to each other. Is it likely that these blocks are linked? In the absence of individual genotypes this argument should be toned down a bit.
13. It would be good to expand on the implications of findings of this study to other systems and organisms. Does lack of high parallelism in other experimental and natural population mean that the role of positive epistasis is not important? Authors have compared their results to other studies which show low parallelism across replicates, are there examples of high parallelism in experimental and natural populations that would support the findings here? And if yes, could positive epistasis also explain those patterns. The authors mention that the pattern they observe here may also be related to the selected trait. Discussing the findings of this paper in a broader context related to adaptation in other organisms/systems would be helpful.
14. In Fig. 1a some of the peaks from the LMM analysis correspond to the identified blocks depicted in the CMH plot with the difference that LMM peaks are narrower than CMH peaks. Considering the extent of linked selection could the selected targets identified by LMM be used to narrow down the selected alleles in haplotype blocks?
15. Please provide an explanation why parallelism for the multiplicative simulations is not affected by the number of loci (Fig. 2).
16. L501-503: Just to clarify does it mean that the populations were maintained in overlapping generations in a laboratory natural selection setting.

Reviewer #3:

Remarks to the Author:

The authors have done an excellent job of addressing my previous concerns. The new long-read reference sequence is particularly helpful for allaying concerns about linkage. The manuscript is well-written, and provides novel and interesting insights into the impact of epistasis in adaptation. I have no further suggestions.

Responses to Reviewers' Comments:

Reviewer comments are in black font. Author responses are in dark blue font.

Reviewer #1 (Remarks to the Author):

1. I think that the authors have done a good job of addressing my concerns from my previous review and am supportive of having this published with minor tweaks. I like their incorporation of explicit modeling of epistasis showing that the results are consistent with these models and less consistent with a broad set of other models. In general, I find their discussion of the results in line with this framing (i.e. positive epistasis can explain the results but has not directly been demonstrated). The one place where I might suggest a change would be the title. Perhaps a more conditional phrase for the role of epistasis should be used, something like 'may be promoted by epistasis'. The addition of new results with chromosome-scale linkage is also very helpful.

Thank you for the supportive remarks. In response to this reviewer's suggestion we have changed the title to "Genome-wide signatures of synergistic epistasis during parallel adaptation in a Baltic Sea copepod". We believe this new title reflects our simulation results while accounting for the reviewer's comment.

I have a few other minor comments as well:

2. line 41-confusing sentence structure and not clear what 'evolutionary response architecture' is.

In response to the reviewer's comment, we have replaced the phrase "evolutionary response architecture" with more commonly used phrases throughout the text, such as "genetic architecture" and "evolutionary genomic response."

Original – "To address this problem, we employed a replicated and controlled evolution experiment using the copepod *Eurytemora affinis* to elucidate the **evolutionary response architecture** to rapid salinity decline, a predicted consequence of global climate change in higher latitudes."

Revised – "To address this problem, we employed an Evolve and resequence (E&R) experiment, using the copepod *Eurytemora affinis*, to elucidate the **evolutionary genomic response** to rapid salinity decline. "

3. line 132 is confusing and may be missing some punctuation

We agree and have adjusted the text as follows:

Original -- To address our goals, we performed pooled whole-genome sequencing at three timepoints during laboratory selection, for ten replicate lines exposed to salinity decline for ten generations and four control lines maintained at constant salinity.

Revised -- To address our goals, we exposed ten replicate lines to salinity decline for ten generations alongside four control lines maintained at constant salinity. These lines were then sampled for pooled whole-genome sequencing at three timepoints.

4. line 156 Fig. 1: Why was this scaffold selected (is it representative or the one with the largest effects)?

This scaffold was selected arbitrarily. This information has been added to the figure legend.

5. Data availability-will the pseudo-reference genome for the European populations be made available in some fashion? Probably would be in some format other than GenBank or I5K

The pseudo-reference genome has now been uploaded to Dryad – temporary link: https://datadryad.org/stash/share/KDsnnngifBRo_wSYedC1IE4kaj3TRRV1RXam-vSH199k.

6. Line 245: It would be good if they at least briefly explain how the results in Fig 2c are obtained and what the terms mean. I may have missed it but didn't find an obvious explanation in methods or experimental methods.

We apologize if this information was difficult to find. The histogram in Figure 2c is simply a different way of displaying the degree of parallelism from the Jaccard index, namely the number of lines in which an allele experienced a significant frequency shift. We have now clarified this in the Figure legend (“The distribution of selected alleles in terms of the proportion of replicate lines in which the selected allele experienced a significant frequency shift at generation ten (i.e. “Replicate frequency spectrum” ref. 41).”) and the information is in the main text line 210 (“... number of replicate lines in which the allele exhibited a significant frequency shift”).

Reviewer #2 (Remarks to the Author):

Authors have made great efforts to incorporate the comments and suggestions of all reviewers. The analysis for the identification of selected alleles has improved and more simulations have been performed. I also very much appreciated embedding figures in the text which made reviewing easier.

Thank you for the positive comments.

1. The role of positive epistasis in the observed high parallelism among replicates relies on the performed simulations and computed summary statistic: Jaccard index. However, I found the description of simulations a bit confusing. I would appreciate it if details of simulations are clarified.

In response to this comment, we moved the detailed descriptions of the model simulations from the supplementary document to the main text and include Table 2 with mathematical details of the simulations.

2. Given the high starting frequency of alleles and strong selection coefficient of the selected alleles, overlap among replicates might also be high under neutrality. It would be interesting to see how the Jaccard index under neutrality is different from the empirical data.

Neutral simulations are already incorporated in our method for detecting selected alleles. We used neutral simulations with empirical starting frequencies to determine an allele frequency change cutoff beyond which an allele was called as under selection in a particular replicate line. This means that neutrality and starting frequency were accounted for when calculating the Jaccard index. Therefore, we would expect that 0.1% (our cutoff) of neutrally simulated alleles would be called as under selection in each replicate line.

Thus, under our framework for detecting selection, it would not be possible to separate out the neutral simulations. However, our simulations did account for the high starting frequencies and selection coefficients of the selected alleles, and the models without epistasis did not match the parallelism of the empirical data (Fig. 2).

3. L629: How is Jaccard index computed for simulated data? Which alleles are considered selected/responding in a replicate in simulations?

We understand that this information may have been difficult to find in the text. As noted in Lines 585-590, alleles with allele frequency changes beyond the 0.1% cutoff from neutral simulations at generation 10 were considered under selection. We have now repeated this information at the place indicated by the reviewer, and also in the Results section on Line 207-209.

4. In Supplementary Fig. 3 why is the empirical Jaccard index lower for SNPs than for blocks? In previous version of the paper with 1156 SNPs, the Jaccard index at generation 10 was around 0.9.

These are excellent questions. The empirical Jaccard index is lower for SNPs than for blocks, simply because there are many more SNPs (4977) than blocks (121), and therefore more noise around which SNPs show signatures of selection in a replicate line. In the previous version of the manuscript where SNPs showed a higher Jaccard index, we used a different set of SNPs, identified as those with both high CMH and LMM scores. Those were likely closer to the targets of selection and showed strong parallel signatures. In this version, we used less stringent significance cutoffs for SNP significance, and calculated parallelism for all the SNPs contributing to a selected haplotype block to make the calculation.

5. I am still surprised that a population set up from 50 individuals with not much recombination can have an estimated N_e of 1750. There should be many uncoupled fertilized eggs and larvae for this to happen. It

seems that N_e is estimated using one method, it would be good to use another method just to make sure more or less similar estimated are obtained.

The number of 50 individuals was not the starting population number in the experiment, but was the number sampled from each replicate line for sequencing.

The number used to establish the laboratory populations was in the thousands. We indicate the number of adults and juveniles used in the starting population on Line 491: “The *E. affinis* copepods used in the laboratory natural selection experiment were collected from Kiel Canal in Kiel, Germany [Lat = 54° 19' 59.88"N, Long = 10° 9' 0"] in 2017 [approximately 1000 copepods] and on May 30, 2018 [85 gravid females and 40 juveniles]”).

These numbers do not account for eggs and larvae, which greatly increases the population size.

Our original estimate of N_e , based on the R package *poolSeq*, was 1750 on average (Supplementary Table 8). Nevertheless, we did estimate N_e using an additional method, WFABC (Foll et al. 2015), which also uses variance in SNP frequency over time to estimate N_e . This method returned an average N_e of 1539 across lines, which is consistent with our original estimate of 1750. These results are now also presented in Supplementary Table 8 and included in the Methods section, Lines 537-544.

6. Supplementary data, section 5: Simulating alleles with starting frequency of 0.5 seems unrealistic even for experimental populations. The effect of starting frequency on parallelism can also be tested by checking whether the high frequency alleles are over-represented as the selected alleles identified in several replicates in simulations using empirical starting allele frequencies.

Simulating alleles with a starting frequency of 0.5 was meant to be an upper bound for the effect of starting frequency on parallelism, not necessarily a realistic scenario (Line 368: “meant to maximize the potential effect”).

Following the recommendation, we tested whether the folded starting frequency was correlated with the number of replicate lines in which the allele was under selection. In the empirical data, we found this correlation to be negative at both generations six and ten (Pearson’s correlation test -- Generation six: $r = -0.260$, t value = -2.94, DF = 119, $P = 0.00399$; Generation ten: $r = -0.214$, t value = -2.40, DF = 119, $P = 0.0183$). This result indicates that alleles with higher starting frequency do not tend to be ‘more parallel’. We have added these results to the text (Lines 361-364).

7. Supplementary data, section 5: ‘Therefore, it is possible that if, for instance, 100 loci contributed to adaptation, all 100 could be located on the same haplotype block or they could all be located on their own haplotype block and be unlinked.’ Would you please clarify whether this means simulations were performed with linkage (or without linkage)?

This means that these particular simulations were performed with linkage among SNPs within haplotype blocks, but the haplotype blocks themselves were unlinked. In response to this comment, we added the following clarifier: “As our SNP simulations considered SNPs on the same haplotype block as linked...”.

8. Parallelism of empirical and positive epistasis model match perfectly in Fig. 2b but not in Fig. S2 in simulations with and without linkage. What is the difference between the simulation parameters of these two figures in terms of number of alleles, starting frequency and s (I assume based on the text that all parameters should be similar to the empirical data)?

We apologize for the confusion. In Supplementary Figure 2, the epistasis simulations used an alpha value of 8, while Fig. 2b shows an alpha of 36.5. The alpha values used in Supplementary Figures 2 and 3 are now explicitly stated in the legend and we have now indicated in every figure legend, as appropriate, that the parameter values can be found in Supplementary Table 3.

9. What is the difference between selective sweep/directional selection and multiplicative model?

The directional selection model is a quantitative genetic model in which allelic effects determine the phenotype, and fitness depends on the phenotypic value and the fitness function. The multiplicative model is a population genetic model in which alleles directly determine fitness in an independent manner. We do not use the term selective sweep in this paper as that term implies the effect of positive selection on linked neutral variation. In response to the reviewer's comments, we have moved the text regarding the details and predictions of these models from the supplementary materials to the main text (Methods -- “Simulation of laboratory selection”; Table 2).

10. In Supplementary Fig. 3 why there is such subtle difference between simulations of empirical starting frequencies and those with starting frequency of 0.5? Is linkage considered in these simulations?

This is an excellent question. The simulations in Supplementary Fig. 3 did include linkage, but the populations started from linkage equilibrium. This is a likely explanation for the subtle difference between models. Modeling linkage at the start of the experiment is outside of the scope of this paper. Not only do we not know the LD structure of the starting population, but it would also be arbitrary and very complicated to simulate under different LD structures and starting frequencies. This point is now noted in the manuscript (Lines 266-267) Future work will hopefully be able to address the effect of LD structure and starting frequency on parallel evolution.

11. The details of simulations which the conclusion of paper heavily relies on are mostly in supplementary files, if there is no page limit from the journal, it would be good to include this important information in the main text.

In response to the reviewer's comments, we have moved the text regarding the details and predictions of these models from the supplementary materials to the main text (Methods -- "Simulation of laboratory selection"; Table 2).

12. The argument about the synergistic effect of ion transporters is interesting and fits well with the positive epistasis notion. However, the GO analysis has been performed using all the SNPs found in the haplotype; this assumes that all these SNPs are true positives. But the motivation for clustering SNPs into blocks is linked selection. The list of ion transporter genes in Table 1 also shows that some of these genes are present on the same blocks. How does the possibility that not all the genes in the selected haplotype blocks might be selected affect the argument about the synergistic effect of ion transporter genes?

This is an interesting point and using all the SNPs contributing to a haplotype block to run the GO analysis could certainly lead to false positives. However, it is clear from Table 1 that the annotated ion transporter genes, which have previously been implicated in freshwater adaptation, are actually on many different haplotype blocks spread throughout the genome. Even if some genes and paralogs are linked, it is plausible that there are synergistic effects among many unlinked ion transporter genes.

13. 339-345: These arguments are based on the assumption that 121 identified haplotype blocks are independent. Just by eyeballing, several of the haplotype blocks (yellow, beige or brownish in the middle of the scaffold) in Fig. 1a seem very close to each other. Is it likely that these blocks are linked? In the absence of individual genotypes this argument should be toned down a bit.

While we agree that some haplotype blocks do appear to be close together in the genome, haplotypes can still be in linkage equilibrium even if overlapping. Furthermore, the aforementioned regions were designated as separate haplotype blocks because SNPs in those regions did not have correlated frequency changes, and were therefore unlikely to be linked.

14. It would be good to expand on the implications of findings of this study to other systems and organisms. Does lack of high parallelism in other experimental and natural population mean that the role of positive epistasis is not important? Authors have compared their results to other studies which show low parallelism across replicates, are there examples of high parallelism in experimental and natural populations that would support the findings here? And if yes, could positive epistasis also explain those patterns. The authors mention that the pattern they observe here may also be related to the selected trait. Discussing the findings of this paper in a broader context related to adaptation in other organisms/systems would be helpful.

Thank you for this excellent recommendation. Some recent experimental evolution studies have reported instances of parallel molecular evolution (e.g. Bitter et al. 2019, Exposito-Alonso et al. 2019, Brennan et al. 2022), but do not compare different models of evolution or quantify the overall degree of parallel evolution, making it difficult to determine whether the degree of parallelism is "high" or "low".

We have now added a few sentences to the Discussion section (Lines 452-453, 458-462) to further discuss our findings in the broader context.:

"It is possible that the low levels of parallelism often reported in wild genomic studies^{2,31-34} could be due, in part, to a lack of power in identifying the targets of selection."

“A few genomic studies of physiological adaptation have reported instances of molecular and phenotypic parallelism, but did not compare support for different mechanisms underlying parallel evolution⁷⁷⁻⁷⁹. Assessing the prevalence of genome-wide epistasis and its impact on parallel evolution across a diversity of traits and systems would provide insights into the role of positive epistasis in promoting rapid and repeated adaptation.”

15. In Fig. 1a some of the peaks from the LMM analysis correspond to the identified blocks depicted in the CMH plot with the difference that LMM peaks are narrower than CMH peaks. Considering the extent of linked selection could the selected targets identified by LMM be used to narrow down the selected alleles in haplotype blocks?

While this is an interesting suggestion, the CMH and LMM tests detect different signatures of selection (i.e., frequency changes greater than drift versus different frequency trajectories between control and selection lines). The difference in peaks could be due, in part, to differing power between the two tests and the different signatures detected. Furthermore, the haplotype block method suggested by the reviewer is already a conservative test that identifies genomic regions under selection rather than a small set of SNPs. Future simulation studies could certainly explore the effects of using different tests to narrow down the selection targets.

16. Please provide an explanation why parallelism for the multiplicative simulations is not affected by the number of loci (Fig. 2).

Under the multiplicative model, allelic effects are totally independent of one another. Therefore, parallelism is only impacted by the allelic effect sizes and population size.

In contrast, for the quantitative trait models the number of loci does affect parallelism because the relative effect sizes of each allele does depend on the number of loci. In other words, each allele has a larger relative effect on the phenotype when there are fewer loci. Therefore, parallelism will increase if the fitness optimum can be reached through changes to a smaller number of (the same) loci.

17. L501-503: Just to clarify does it mean that the populations were maintained in overlapping generations in a laboratory natural selection setting.

Yes, this does mean they were maintained in overlapping generations in a laboratory natural selection setting. We have added this point to the text in Lines 505-506:

“Animals were not transferred to the next generation but instead were allowed to survive and reproduce undisturbed with overlapping generations.”

Reviewer #3 (Remarks to the Author):

The authors have done an excellent job of addressing my previous concerns. The new long-read reference

sequence is particularly helpful for allaying concerns about linkage. The manuscript is well-written, and provides novel and interesting insights into the impact of epistasis in adaptation. I have no further suggestions.

Thank you for the positive remarks!

Reviewers' Comments:

Reviewer #2:

Remarks to the Author:

Authors have addressed/answered all my comments/questions. I also appreciate the change of title which now reflects the findings of the manuscript better. It's been great to see the evolution of this manuscript since the first version, in particular including simulations that specifically model epistasis has improved the manuscript. I have no further suggestions.